# Molecular architecture and electron transfer pathway of the Stn family transhydrogenase

Anuj Kumar[1,2,5], Florian Kremp[2,5], Jennifer Roth[2], Sven A. Freibert [1,3,4], Volker Müller [2] ✉ & Jan M. Schuller [1] ✉

The challenge of endergonic reduction of NADP+ using NADH is overcome by ferredoxin-dependent transhydrogenases that employ electron bifurcation for electron carrier adjustments in the ancient Wood-Ljungdahl pathway. Recently, an electron-bifurcating transhydrogenase with subunit compositions distinct from the well-characterized Nfn-type transhydrogenase was described: the Stn complex. Here, we present the single-particle cryo-EM structure of the Stn family transhydrogenase from the acetogenic bacterium *Sporomusa ovata* and functionally dissect its electron transfer pathway. Stn forms a tetramer consisting of functional heterotrimeric StnABC complexes. Our findings demonstrate that the StnAB subunits assume the structural and functional role of a bifurcating module, homologous to the HydBC core of the electron-bifurcating HydABC complex. Moreover, StnC contains a NuoG-like domain and a GltD-like NADPH binding domain that resembles the NfnB subunit of the NfnAB complex. However, in contrast to NfnB, StnC lost the ability to bifurcate electrons. Structural comparison allows us to describe how the same fold on one hand evolved bifurcation activity on its own while on the other hand combined with an associated bifurcating module, exemplifying modular evolution in anaerobic metabolism to produce activities critical for survival at the thermodynamic limit of life.

Life on earth requires the constant input of energy to produce and maintain cellular matter. Energy is conserved by cells from redox reactions by respiration or fermentation with the ultimate goal to produce the universal cellular energy currency, adenosine triphosphate (ATP). Hydrolysis of ATP then enables the endergonic synthesis of cellular constituents such as proteins, nucleic acids, and lipids from either $CO_2$ in autotrophs or smaller organic building blocks in heterotrophs. In addition to ATP, electron flow is required to link the catabolic with the anabolic metabolism pathways[1]. In both biochemical pathways, nicotinamide adenine dinucleotide (NADH) or nicotinamide adenine dinucleotide phosphate (NADPH) serve as electron carriers. Whereas NADH is the usual carrier in catabolic reactions, NADPH

serves as electron donor in most anabolic reactions. The biochemical connection between the pools of NADH and NADPH is catalysed by transhydrogenases existing in all domains of life. Two types of transhydrogenase are known in aerobic organisms: one membrane-bound, proton-pumping NADH:NADP+ oxidoreductase (PntAB), and one soluble form of transhydrogenase (UdhA) that is independent from the proton gradient $\Delta\bar{\mu}_{H}^{+}$ across the membrane[2,3]. The existence of UdhA is restricted to prokaryotes[4], whereas PntAB can also be found in the inner membrane of mitochondria in eukaryotes[5]. Although the standard redox potentials ($E^{0'}$) of the NAD+/NADH and NADP+/NADPH pairs are both −320 mV, within the cell the ratio of NAD+:NADH is 30:1, whereas the ratio of NADP+:NADPH is 1:40, which results in redox

[1]SYNMIKRO Research Center and Department of Chemistry, Philipps-University of Marburg, Marburg, Germany. [2]Department of Molecular Microbiology & Bioenergetics, Institute of Molecular Biosciences, Johann Wolfgang Goethe University, Frankfurt am Main, Germany. [3]Institut für Zytobiologie im Zentrum SYNMIKRO, Philipps-University of Marburg, Marburg, Germany. [4]Core Facility "Protein Biochemistry and Spectroscopy", Marburg 35032, Germany. [5]These authors contributed equally: Anuj Kumar, Florian Kremp. ✉e-mail: vmueller@bio.uni-frankfurt.de; jan.schuller@synmikro.uni-marburg.de

potentials (E') of −280 mV and −370 mV, respectively[6]. Thus, hydride transfer from NADH to NADP$^+$ is highly endergonic and the energy required to drive this reaction is provided by proton influx, catalyzed by PntAB. In contrast, the soluble UdhA is only able to catalyse the exergonic reduction of NAD$^+$ with NADPH[3,7], but not the endergonic back reaction.

Interestingly, unlike aerobic organisms, anaerobes neither have PntAB nor UdhA, which is why the connection between the pyridine nucleotide pools was obscure for a long time. In 2010, the missing link was found: a completely reversible, NADH-dependent Fd:NADP$^+$ oxidoreductase (Nfn) was discovered by Wang et al.[8] in *Clostridium kluyveri*. The Nfn complex is soluble and able to run the endergonic reduction of NADP$^+$ with NADH by using ferredoxin-dependent NADP$^+$ reduction as driving force via flavin-based electron bifurcation (FBEB)[8]. FBEB is a mechanism discovered in 2008 to couple an exergonic with an endergonic redox reaction in one soluble enzyme complex[9]. In general, one electron donor is oxidised with the concomitant reduction of two different electron acceptors, of which one is usually the low potential electron carrier ferredoxin. In Nfn, exergonic electron transfer from NADPH to NAD$^+$ drives endergonic electron transfer from NADPH to ferredoxin. Vice versa, endergonic electron transfer from NADH to NADP$^+$ is catalysed by ferredoxin (Fd) as reductant in the exergonic reaction. Central to the electron bifurcation/confurcation is the flavin cofactor: flavins can act as one- or two-electron carriers. In the special flavin cofactor of FBEB enzymes the reduced flavohydroquinone transfers one electron after the other, thereby the formation of a short-lived anionic semiquinone (ASQ) intermediate is favoured by the protein environment provided by the enzyme. This results in crossed potentials of the hydroquinone/anionic semiquinone (HQ/ASQ) and the anionic semiquinone/oxidised quinone (ASQ/OQ) couple, meaning that the ASQ/OQ has a more negative redox potential than the HQ/ASQ. This allows one-electron transfers to acceptors with high and low potentials[10]. In addition to the bifurcating flavin, a second flavin is usually required to collect the single electrons before a hydride can be transferred to NAD$^+$ or NADP$^+$.

Nfn is an enzyme complex consisting of only two subunits, NfnA and NfnB. NfnB harbours two [4Fe4S]-clusters, the bifurcating flavin adenine dinucleotide (b-FAD), and the NADP(H) and ferredoxin binding sites[11]. NfnA has the NAD(H) binding site and binds to a [2Fe2S]-cluster and a second FAD molecule (a-FAD). In the bifurcation mode, two electrons from NADPH are transferred to the bifurcating FAD cofactor[11]. One electron travels via the [2Fe2S]-cluster to the a-FAD in NfnA, leaving a low potential FADH•- radical in NfnB that allows the reduction of ferredoxin via the two [4Fe4S]-clusters of NfnB. After a second cycle, the high potential electron acceptor NAD$^+$ is then reduced in a two-electron transfer step from NfnA-bound FADH$^-$. However, many anaerobes do not rely on Nfn for balancing the nucleotide pool. Very recently, a different transhydrogenase (Stn) was discovered in the strictly anaerobic acetogenic bacterium *Sporomusa ovata*[12]. Stn is also widely distributed in anaerobic bacteria and most species either have genes for Nfn or Stn. Interestingly, Stn has three subunits, two of which that are evolutionary unrelated to Nfn but share structural identity with the subunits HydB and C of the electron-bifurcating [FeFe]-hydrogenase HydABC[12–14].

In this work, we have determined the single-particle cryo-electron microscope (cryo-EM) structure of the StnABC complex of *S. ovata* at 3.0 Å resolution under catalytic turnover conditions. Additional structure-based, site-directed mutagenesis experiments further allowed characterizing the pathways for electron transport and the electron bifurcation mechanism of the hitherto uncharacterized transhydrogenase complex.

## Results

### Purification of the StnABC complex

The StnABC complex was purified to apparent homogeneity from *S. ovata* as previously described[12] (Supplementary Fig. 1a, b). The complex was highly active with physiological and artificial electron donors: it catalysed the Fd$_{red}$ and NADH-dependent reduction of NADP$^+$ with an activity of 5.57 U/mg, the Fd$_{ox}$-dependent reduction of NAD$^+$ with NADPH with an activity of 2.80 U/mg, and the NAD$^+$-dependent reduction of Fd$_{ox}$ with NADPH with an activity of 1.59 U/mg (Supplementary Fig. 1c, Supplementary Table 1). In electron bifurcating enzymes the electron acceptors are reduced in a ratio of 1:1. For the StnABC a deviation from the 1:1 ratio is known[12], which can be explained by the fact that the Stn complex not only catalyses the Fd$_{red}$:NADP$^+$ oxidoreductase activity, which pushes the ratio towards NAD$^+$ reduction, but also a NADPH:NAD$^+$ oxidoreductase activity, which does the same[12]. The reduction of methyl viologen (MV) with NADPH was also catalysed with an activity of 164.08 U/mg (Supplementary Table 1). For site-directed mutagenesis experiments, a His-tagged complex was produced heterologously in *E. coli*. The wildtype (WT) StnABC-His catalysed the reduction of NAD$^+$ and Fd$_{ox}$ with NADPH with an activity of 1.08 and 0.72 U/mg, respectively, the NADH- and Fd$_{red}$-dependent reduction of NADP$^+$ was catalysed with an activity of 1.35 U/mg, and the reduction of MV$_{ox}$ with NADPH was catalysed with 86.37 U/mg (Supplementary Table 1). The purified complex of the StnABC-His contained 42.6 ± 1.4 mol iron/mol StnABC, which corresponds well with the number of Fe expected (42), and FAD as a cofactor. None of the below-mentioned variations led to a loss of flavin in the enzyme. Along with the StnABC-His complex, traces of impurities were also detected after gel filtration that were identified by MALDI-TOF analysis as *E. coli* proteins (Supplementary Fig. 1d, e).

### The molecular architecture of StnABC

To reveal the molecular structure of StnABC from *S. ovata*, we subjected the protein complex to single-particle cryo-EM. All cryo-EM grids were prepared in a strictly anoxic chamber with a gas composition of 95% N$_2$ and 5% H$_2$. We determined the structure of the StnABC complex in two states. One state is the isolated or purified state of the enzyme in the absence of any added substrates (NADPH, NAD$^+$, and Fd) and cofactor (FMN), referred to as StnABC$_{S1}$ state hereafter (Supplementary Fig. 2). The second state is the purified enzyme with the addition of exogenous cofactor (FMN) and substrates (NADPH, NAD$^+$, and Fd), referred to as StnABC$_{S2}$ state hereafter (Supplementary Fig. 3). The NADPH acts as a natural reducing agent of the enzyme to reduce the bound cofactors. Of note, the cofactor FAD was added to both states during purification (see Methods).

At the level of reference-free 2D class averages the dihedral (D2) symmetry of the molecule is apparent, representing a tetramer as a whole (Supplementary Fig. 2b and Supplementary Fig. 3b). The central core of the complex was found to be rigid, although the peripheral parts exhibited diffusive and weaker electron density, indicating the presence of more flexible parts in the complex (Supplementary Fig. 4). After 2D classification, the best particles were used for 3D reconstruction with applied D2 symmetry, where they were refined to obtain a global resolution of 3.2 Å for the StnABC$_{S1}$ and 3.0 Å for the StnABC$_{S2}$ state of the complex (Fig. 1a, Supplementary Figs. 2–4, Supplementary Table 2). Both structures show the same overall architecture and conformational landscape. However, they differ in the occupancy of substrates and cofactors. In the structure of the StnABC$_{S1}$ state, only FAD was present in the StnC subunit but no FMN could be observed in the binding cavity of the StnB subunit. Since the cryo-EM structures of the StnABC$_{S1}$ and StnABC$_{S2}$ states are similar, we focus our description on the StnABC$_{S2}$ because it harbours all cofactors or prosthetic groups ([4Fe4S] and [2Fe2S]-clusters, Zn$^{2+}$ ion, FAD, and FMN) (Supplementary Table 3), including the NAD$^+$ and NADPH in the complex (Fig. 2a, b, Supplementary Figs. 3, 4, Supplementary Tables 2, 3, Supplementary Movie 1). Unfortunately, capturing Fd was not feasible, likely due to its transient binding.

The density allowed a de novo model building of the majority of the StnABC complex and AlphaFold[15] was used for constructing the

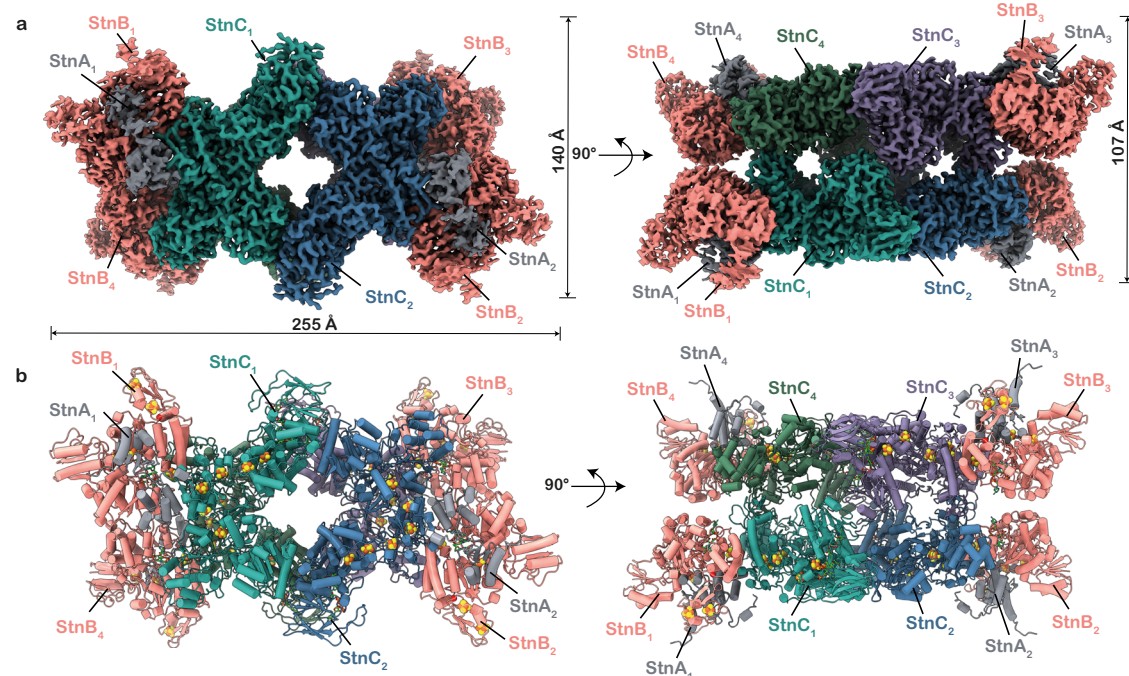

**Fig. 1 | Molecular architecture of StnABC. a** Three-dimensional segmented cryo-EM density of the tetrameric complex coloured by subunits. **b** Corresponding views of the StnABC atomic model in cartoon representation. The tetramer is mediated by the interactions of StnC subunits.

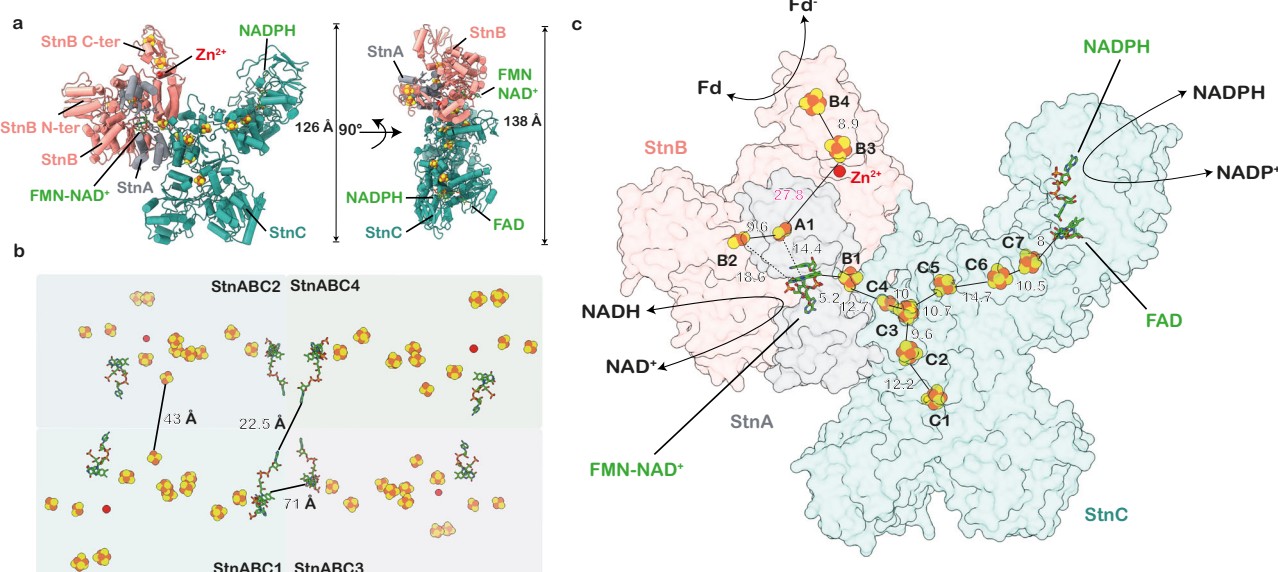

**Fig. 2 | Cofactor organization in StnABC complex. a** Two views of StnABC functional protomer. StnA contains one [2Fe2S]-cluster; StnB contains FMN and NAD$^+$ cofactors, including three [4Fe4S]-clusters, one [2Fe2S]-cluster, and a zinc atom; StnC harbours six [4Fe4S]-clusters and one [2Fe2S]-cluster with FAD and NADPH cofactors. **b** The overall cofactors arrangement in StnABC tetrameric complex with distances between the neighbouring protomers marked. The distances are outside the physiological range of electron transfer between each protomer. **c** Overview of cofactor organisation and the edge-to-edge distances in Ångströms of StnABC complex. An electron transfer path is indicated from the FAD-NADPH site in the StnC to FMN in the StnB subunit. The reduction of NAD$^+$ takes place at FMN, and Fd is reduced at the C-terminus of StnB.

initial model (Fig. 1b). To address the flexibility of the outer regions, namely the N- and C-terminus of StnB and the C-terminus of StnA, we refrained from refining the orientations of the side chains. Instead, we utilized secondary structure information by docking the AlphaFold[15] models into the electron density of those regions (Supplementary Fig. 4). The overall dimensions of the resulting structural model are 255 × 140 × 107 Å (Fig. 1a). The observed complex consists of four protomers, each protomer being a StnABC heterotrimer, wherein the

tetrameric core is composed of the StnC subunit from each of the four protomers (Fig. 1a, b). Individual StnC subunits in the tetramer are flanked by a more dynamic StnAB subcomplex (Figs. 1a, b, 2a), with only StnB bound to StnC, whereas StnA was found to be interacting with StnB through multiple contact points (Fig. 2a). The overall complex comprises a large number of redox cofactors: StnC has six [4Fe4S]-clusters and one [2Fe2S]-cluster (C1-C7) as well as bound FAD and NADPH molecules. The StnB subunit has one [2Fe2S]-cluster (B2),

three [4Fe4S]-clusters (B1, B3 and B4), and a solvent-accessible cavity harbouring bound FMN and NAD+ in a stacking conformation (Fig. 2c). StnA consists of a single binuclear [2Fe2S]-cluster (A1) in its C-terminal thioredoxin-like domain. Apparent from the structure, the average edge-to-edge distances of all the iron-sulphur clusters ranged between 9–18 Å in a single protomer, ensuring rapid electron transfer across the entire structure[16] (Fig. 2c). A bottleneck distance of more than 25 Å was observed between the A1 and B3 clusters of the StnA and StnB subunits, respectively (Fig. 2c). The distances between the nearest cofactors of individual protomers were outside the regime for physiological electron transfer (71, 43, and 23 Å) (Fig. 2b), which is in agreement with the hypothesis that a single protomer represents the minimal functional unit.

It has been reported previously[12] that the StnABC complex only harbours FAD and still performs the bifurcation assay without the addition of exogenous FMN. Therefore, during the initial stages of determining the structure of the StnABC complex, only FAD was added to our protein preparations (see Methods). As expected, the StnABC$_{S1}$ state of the enzyme was found to contain only FAD in StnC subunit (Supplementary Tables 2 and 3), whereas the StnB subunit lacked an FMN despite being homologous to the HydB from the HydABC complex (Supplementary Fig. 5c). We reasoned that FMN is only loosely bound in StnB and only a fraction of the protein complex harboured FMN during the protein purification. Therefore, the electron density for FMN was averaged out within the density for the StnABC$_{S1}$ structure. The phenomenon of loosely bound FMN molecules was also reported for the electron bifurcating hydrogenase HydABC, which displays increased bifurcating activity when exogenous FMN is provided[13], suggesting that activity displayed in the absence of exogenous FMN is attributable to a fraction of the population that retains FMN and got co-purified with the protein, despite modest binding affinity. Further, studies have shown that the addition of exogenous FMN yields higher bifurcation activity for StnABC and HydABC complexes[12,13,17,18] as more protein complexes can bind FMN. To get a more comprehensive view of which flavin could be involved in electron bifurcation, we analysed the structure of StnABC in the presence of exogenous FMN, NAD+, Fd, and NADPH. The StnABC$_{S2}$ structure demonstrated the full occupancy of FMN·NAD+ in all StnB subunits of the tetrameric assembly. The comparison between both states (StnABC$_{S1}$ and StnABC$_{S2}$) provided preliminary evidence that the StnABC complex could perform bifurcation similar to HydABC complexes.

## StnC shuttles electrons from NADPH to the StnAB bifurcating module

StnC is a large protein of 1172 amino acids assembled from the fusion of the NADPH binding Glutamate synthase (GltD) and NADH:ubiquinone oxidoreductase subunit G (NuoG) domains. The N-terminal region (amino acids 102–565) closely resembles a nucleotide-binding fold of GltD. Furthermore, the region displays a similarity to the NfnB subunit (also known as NfnL) from *Pyrococcus furiosus* with an RMSD of 1.2 Å (Supplementary Fig. 5a, d). It also harbours two [4Fe4S]-clusters (C7 and C6) and a binding site for FAD and NADPH molecules. The NuoG domain of the StnC subunit contains four [4Fe4S]-clusters and one [2Fe2S]-cluster (C1-C5) (Supplementary Fig. 5a, d). The NuoG domain is a blend of a [Fe]-only hydrogenase (amino acids 1–101) containing a [2Fe2S]-cluster (C4) and a molybdopterin enzyme-like formate dehydrogenase (FdhF, amino acids 683–1172). The C-terminal domain within the NuoG domain of StnC contains two [4Fe4S]-clusters (C1 and C2) and is closest to FdhF from *Desulfovibrio gigas* with a corresponding RMSD value of 3.2 Å (Supplementary Fig. 5d); however, the FdhF domain is catalytically inactive. The tetramer formation is exclusively mediated by the StnC subunits via multiple contact points established between the FdhF-like and the N-terminal domains. There is a linear electron transfer pathway from the FAD-NADPH binding site

in the StnC subunit to the StnAB subcomplex via C7, C6, C5, C4, and C3 clusters. The C1 and C2 clusters are not a part of this pathway as this branch is a "non-functional" evolutionary remnant in terms of electron transfer (Figs. 2c, 3a).

To identify the function of C1 and C2 we further analysed the sequences and alphafold models of two of the closest homologs of StnC, the NADPH-dependent sulfur oxidoreductase (NsoC) from *Thermococcus litoralis*[19] and the NfnA from *Thermococcus sibiricus*[20] (Supplementary Fig. 6a, b). Compared to the FdhF-like domain of StnC, the NsoC and NfnA are predicted to have partial C-terminal domain with a lack of cysteines required for the binding of cluster C2 (Supplementary Fig. 6b), underlining the notion that C1 and C2 are not involved in electron transfer. In addition, to probe the functional role of C1 and C2 clusters, StnABC variants were heterologously produced in *E. coli*, purified, and their enzymatic activities were determined (Supplementary Fig. 7b, c). Deletion of the FdhF domain in StnC that contains the clusters C1 and C2 resulted in a failure to purify a functionally active StnABC-His complex (Supplementary Fig. 7a–c). Also, the exchange of the cysteines that bind the [4Fe4S]-cluster C1 or C2 with alanine resulted in a failure in tetramer formation (Supplementary Fig. 7b, c). Mass photometry revealed that the subunits did not form correct assemblies (Supplementary Fig. 7d). We, therefore, conclude that the branching [4Fe4S]-clusters and the FdhF domain in the StnC subunit seem to play an important role either in the stability of the complex or the initial assembly of the tetramer. However, the exact role of C1 and C2 clusters is not clearly understood.

## FAD in StnC is the non-bifurcating flavin

The FAD-NADPH binding site in StnC is homologous to the one in NfnB of the electron-bifurcating NfnAB complex. As observed in NfnB, the nucleotides bind to two different Rossmann folds and the important residues coordinating the FAD cofactor are conserved in both proteins (Fig. 3a, c–e, Supplementary Fig. 8d). In NfnB, R201 appears to form hydrogen bond with the N5 nitrogen of FAD, thereby protecting the N5 from protonation, a condition that is considered crucial for efficient electron bifurcation[21,22] (Fig. 3e). Since N5 cannot be protonated, a high-energy (low-potential) anionic semiquinone (ASQ) is formed instead of a neutral semiquinone (NSQ)[22]. The corresponding arginine residue in StnC (R239) is not involved in forming a hydrogen bond with the N5 nitrogen of FAD due to its long distance of ~6 Å (Fig. 3c)[23]. Instead, this residue is essential for maintaining the overall activity of the StnABC complex. The fully intact, heterologously produced StnABC complex containing the variants StnC_R239K and StnC_R239A did catalyse the simultaneous reduction of NAD+ and Fd$_{ox}$ with NADPH (Fig. 3f, Supplementary Fig. 8a, b, Supplementary Table 1), but with significantly lower activities (0.33 U/mg and 0.03 U/mg, respectively) compared to the WT StnABC-His enzyme (Fig. 3f and Supplementary Table 1). The Fe contents for both variants were found to be at WT level: 41.8 ± 1.7 for StnC_R239K and 41.7 ± 2.8 for StnC_R239A. Also, the NADH- and Fd$_{red}$-dependent reduction of NADP+ was catalysed by StnC_R239K with an activity of 0.63 U/mg, but the activity of StnC_R239A was below the detection limit (Supplementary Table 1). Rates for reduction of MV$_{ox}$ with NADPH as catalysed by StnC_R239K and StnC_R239A were lower compared to the WT StnABC-His enzyme, but with 65.42 U/mg and 5.38 U/mg, respectively, they were still rather high (Supplementary Table 1). Given the lack of coordination of R239 to N5 of FAD in StnC, we hypothesize that FAD does not form an ASQ as required for the bifurcation reaction, but rather forms an NSQ. Upon hydride transfer from NADPH to FAD, the electrons are sequentially delivered to the FMN in StnB via a chain of [4Fe4S]-clusters (C7, C6, C5, C4, C3, B1) in StnC and StnB subunits.

The [4Fe4S]-cluster C7 next to FAD has an unusual ligand sphere where three cysteine residues along with a lysine residue (K170) seem to coordinate the metals in the [4Fe4S]-cluster in the StnABC$_{S1}$ state of the enzyme (Fig. 3b). Astonishingly, in the StnABC$_{S2}$ state with

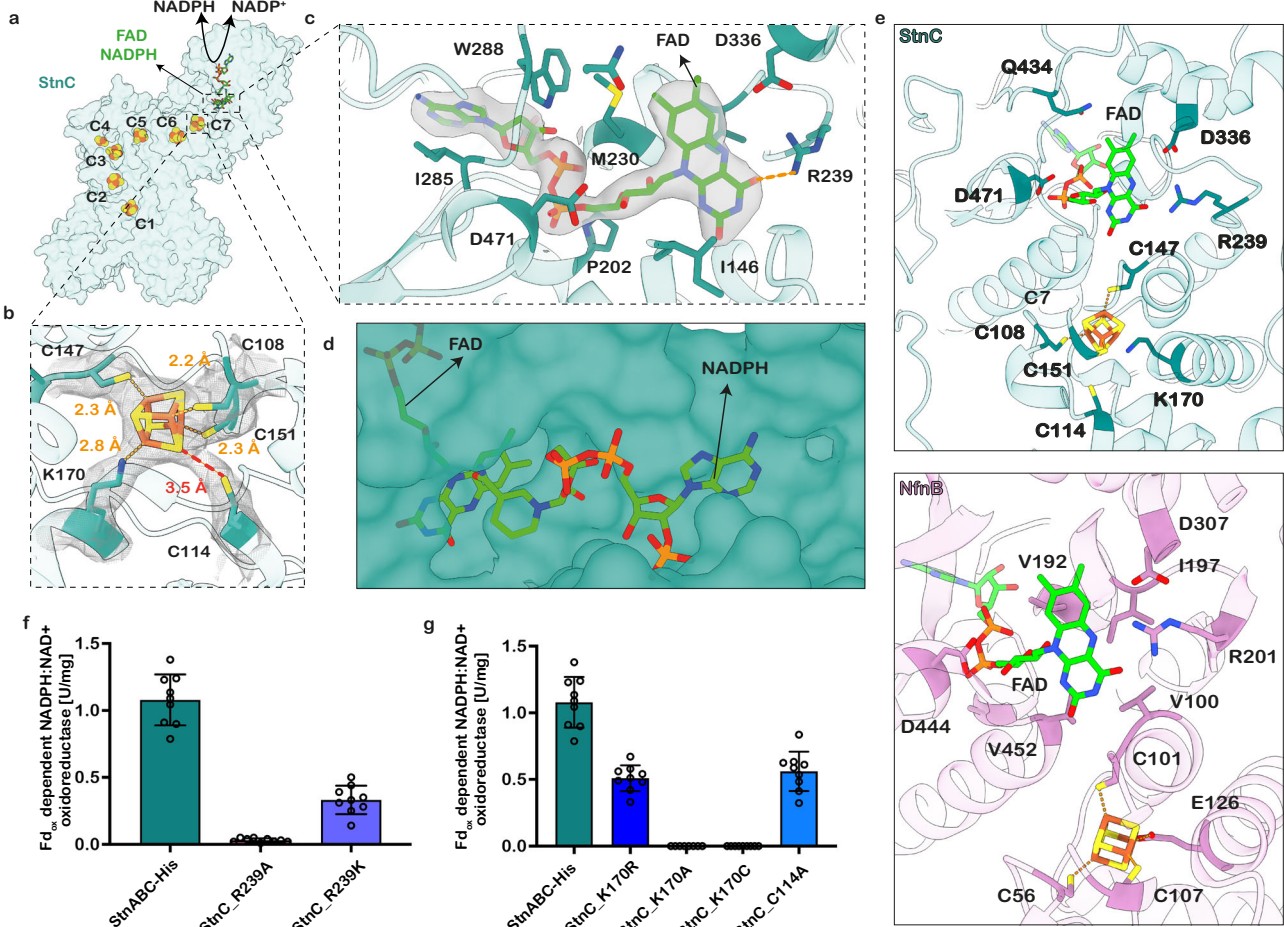

**Fig. 3 | Mutational analysis of the StnC subunit. a** Surface representation of StnC with six [4Fe4S]-clusters and one [2Fe2S]-cluster, along with the FAD-NADPH binding site shown. The oxidation of NADPH triggers the electron flow from FAD in StnC to the FMN in StnB. **b, c** Zoom-ins of the FAD binding site and C7 cluster encased around cryo-EM density. **d** NADPH binds at the pocket carved by two Rossmann folds in the StnC subunit. **e** Comparison of the FAD binding site in StnC and NfnB subunits. Residue R201 in NfnB forms a hydrogen bond with N5 of FAD. The corresponding R239 in StnC also participates in hydrogen bonding but not with the N5 of FAD. **f, g** Transhydrogenase activity assays performed for different complex variants probing the roles of R239, K170, and C114. $Fd_{ox}$-dependent NADPH:NAD$^+$ oxidoreductase activity of StnABC-His in comparison to variants StnC_R239A, StnC_R239K, StnC_K170R, StnC_K170A, StnC_K170C and StnC_C114A. Activities are the mean ± s.e.m. of three independent biological replicates, measured in triplicates (*n* = 3). Activities are given in U/mg. One Unit is defined as the transfer of 2 μmol electrons/min.

FMN-NAD$^+$ bound, the lysine coordination largely disappeared and instead a strong new density could be observed between Cys114 and acid-labile sulphur of the C7 (Supplementary Fig. 9).

In NfnB, the lysine is replaced by a glutamate residue (Fig. 3e, Supplementary Fig. 8d). Mutation of K170 to alanine or cysteine resulted in a severe loss of physiological activities, whereas an arginine variant retained 50% of its WT activity (Fig. 3g, Supplementary Table 1). This indicates that a residue that can be deprotonated to provide a lone electron pair is essential at this position. Comparably, the NADPH:$MV_{ox}$ oxidoreductase activity was also decreased by ~50% (for variant K170R) compared to the StnABC-His, whereas the variants K170A and K170C catalysed the $MV_{ox}$ reduction with NADPH with only 9.35 and 9.54 U/mg, respectively, although the Fe content in all three variants was at WT-level (41.5 ± 2.0) (Supplementary Table 1). We speculate that the reduced activity might result from an altered redox potential of the C7 cluster in StnABC_K170A and K170C, which ultimately inhibited electron transfer because the distance between the A170 or C170 and the iron ion of the [4Fe4S]-cluster is too large to stabilize the cluster. Arginine, on the other hand, resembles lysine in length and properties and might be able to functionally replace lysine.

The overall importance of the residue K170 in StnC is further supported by the finding that K170 is strictly conserved in all potential StnC sequences identified so far[12] (Supplementary Fig. 8c). In our structure the residue C114 could potentially coordinate C7; despite the distance between them being too large (3.5 Å) for direct coordination (Fig. 3b), we have identified a strong connecting cryo-EM density in the StnABC$_{S2}$ state of the enzyme (Supplementary Fig. 9). The mutation of this cysteine to an alanine did not change the Fe content of Stn, but reduced the activity by 50% (Fig. 3g, Supplementary Table 1). This suggests that C114 indeed plays a role in either modulating the exact redox potential or creating the properly folded environment of this unusual cluster. Unlike K170, C114 seems not strictly conserved among the StnC homologs but is replaced by valine or isoleucine in two cases (Supplementary Fig. 6a, Supplementary Fig. 8c). However, little is known about the actual functionality of these homologues. In summary, our observations provide strong evidence that both K170 and C114 play important roles in stabilizing and modulating the [4Fe4S]-cluster C7. Functionally, these residues may adjust and modify the redox potential of C7 in a manner similar to the non-traditional cluster found in NfnB. This could promote fast electron transfer from the FAD to the FMN site in StnB, although experimental verification is pending.

## The StnAB module is responsible for the electron bifurcation in the complex

The StnAB subunits form the electron-bifurcating core of the enzyme complex that reversibly reduces $NAD^+$ and Fd (Fig. 4a, b). Owing to high similarity with the HydBC core of the electron-bifurcating hydrogenase HydABC from *Thermoanaerobacter kivui* (RMSD value of 1.17 Å) (Fig. 4f, Supplementary Fig. 5b, c), it can be reasoned that StnAB might use a similar mechanism to perform the bifurcation reaction[17]. Indeed, similar to HydB, StnB carries the hallmark features of the electron bifurcation module of the HydABC complex with the N-terminal thioredoxin-like domain and a C-terminal bacterial Fd-like domain representing a putative binding site for Fd reduction (Supplementary Fig. 5a, c). This domain is connected via a linker to the core of StnB, where the linker is stabilized by a bound $Zn^{2+}$ ion (Figs. 2a, c, 4a, f, Supplementary Fig. 4b). The $Zn^{2+}$ ion was identified by ICP−MS and was indicated by a strong electron density at this region in our cryo-EM structures (Supplementary Fig. 4b). In the vicinity of the FMN binding site, the C-terminal thioredoxin-like domain of StnA forms multiple contact points with the Rossmann and four helical-bundle domains of StnB (Fig. 4b, Supplementary Fig. 5a, c). This allows the StnA subunit to bring the [2Fe2S]-cluster A1

in close proximity to the FMN and the B2 cluster in the StnB subunit (Figs. 2c, 4b). This arrangement forms a unique iron-sulphur cluster environment, which may allow a single FMN to gate the exergonic and endergonic reduction of $NAD^+$ and Fd, respectively, as recently shown for the HydABC complexes from different organisms[17,18,24]

The FMN binds in a solvent-accessible cavity of the StnB core with a glycine-rich loop (Fig. 4b, c). In our cryo-EM structure of the $StnABC_{S2}$ state, we find full occupancy of FMN with a bound $NAD^+$ nucleotide (Fig. 4b). The $NAD^+$ binding site with FMN is fairly similar to that in HydB of the electron-bifurcating hydrogenase[17], where the recognition, binding, and release of the $NAD^+$ is coordinated by three phenylalanine residues. The $NAD^+$ forms a stacking interaction with FMN, placing the aromatic rings at a distance of 4 Å from each other, which allows a complete two-electron reduction of $NAD^+$ (Fig. 4b, c). The comparison of StnABC structures in both the states revealed no major differences; however, we could observe the loop (residues 182−191) of StnB undergoing a moderate conformational change upon $NAD^+$ binding (Fig. 4e). Here, the loop dynamics is primarily due to the binding of $NAD^+$ and not simply due to FMN, which has been shown previously by Katsyv and co-workers[17]. The loop comprising an aspartate residue (D186) could presumably be

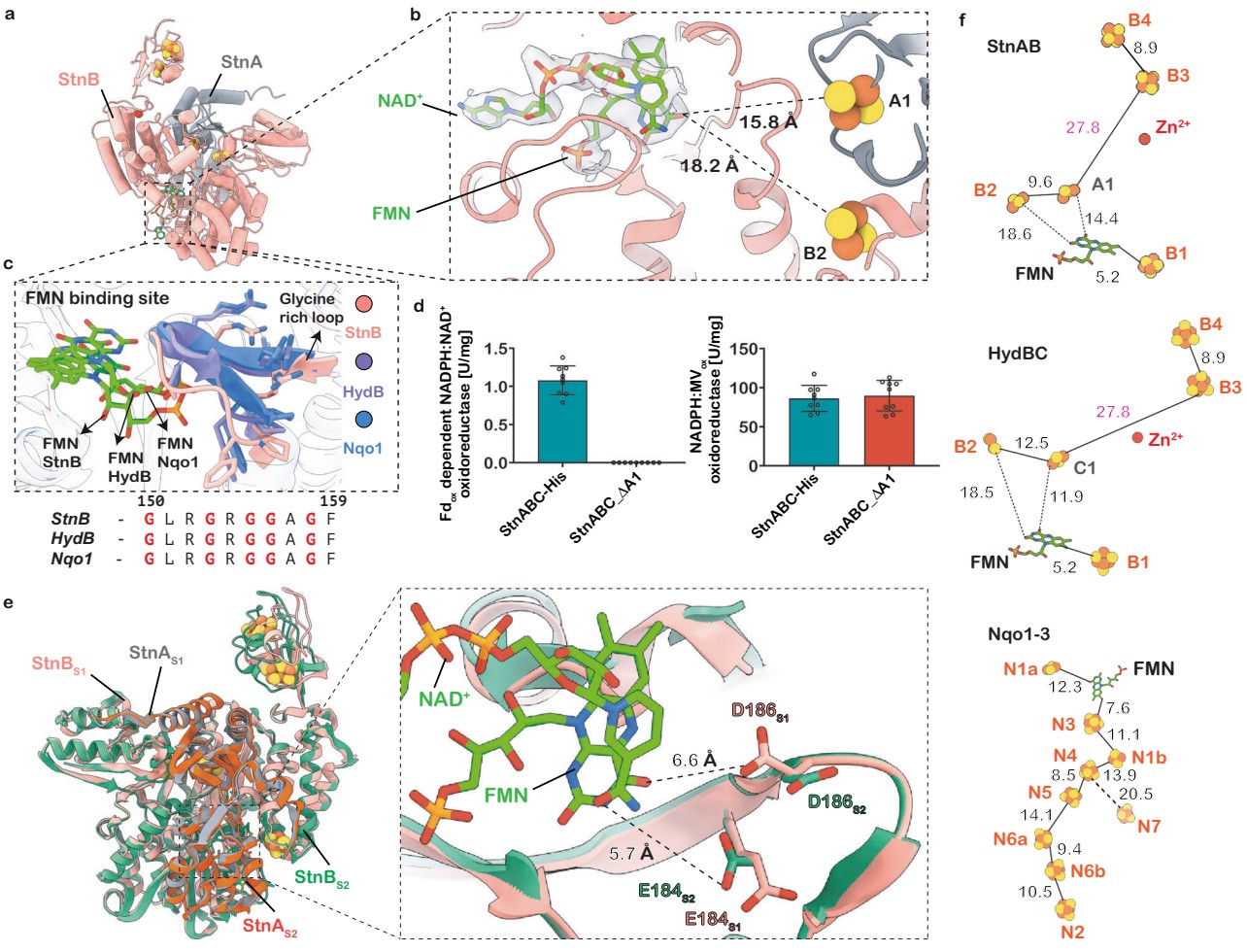

**Fig. 4 | FMN-NAD⁺ binding and mutational analysis of the electron bifurcating StnAB module. a, b** Structural model of StnAB subunit, with the corresponding cryo-EM density enveloping the FMN-NAD⁺ pair. The distances of nearby A1 and B2 clusters to the FMN are shown. **c** Superposition of FMN binding sites in StnB, HydB, and Nqo1. All three subunits harbour and stabilize the bound FMN using the conserved glycine-rich loop. **d** Comparison of the NADPH:MV$_{ox}$ oxidoreductase activity and Fd$_{ox}$-dependent NADPH:NAD⁺ oxidoreductase activity of StnABC-His and variant StnABC_ΔA1. Activities are the mean ± s.e.m. of three independent

biological replicates, measured in triplicates ($n = 3$). Activities are given in U/mg. One Unit is defined as the transfer of 2 μmol electrons/min. **e** Overlay of the StnAB module structure determined under the $StnABC_{S1}$ and $StnABC_{S2}$ state of the enzyme. The loop (residue 182−191) undergoes a slight conformational change upon the binding of NAD⁺ in the $StnABC_{S2}$ state of the enzyme. **f** Comparison of the electron transfer path in the StnAB module with HydBC and Nqo1-3 subunits. In all three cases, the FMN acts as a mediator to transfer the electrons sequentially to nearby clusters. All the distances given are in Ångströms (Å).

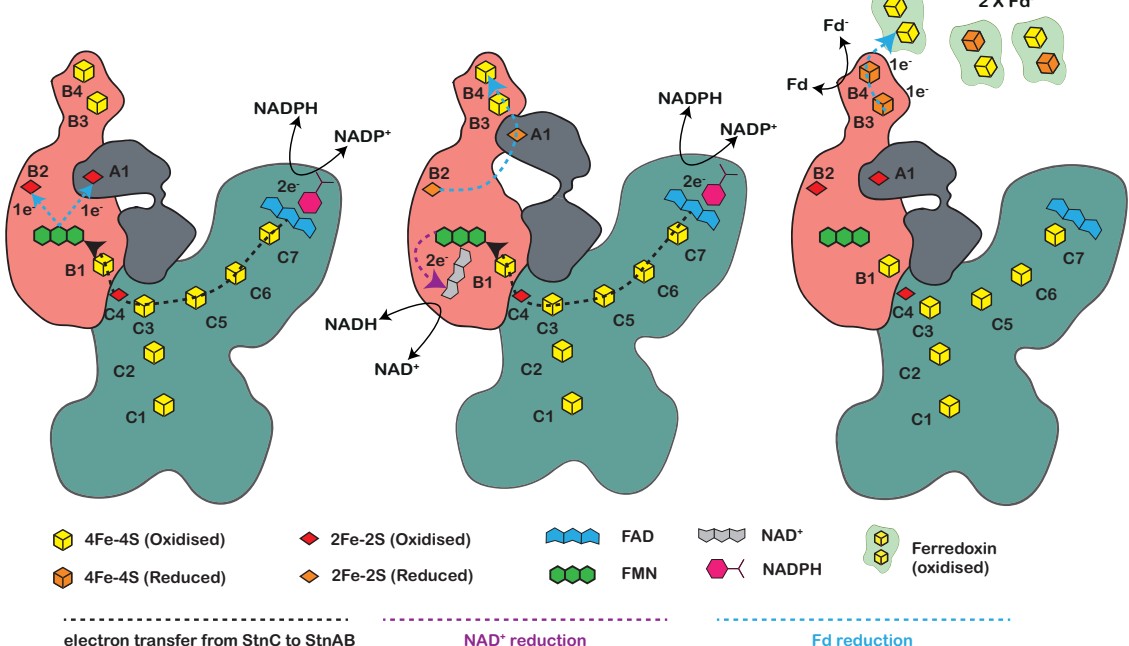

**Fig. 5 | Putative mechanism of electron bifurcation in the StnABC transhydrogenase.** The reaction is initiated after NADPH binding and its oxidation at the FAD cofactor in the GltD domain of StnC. During the first round of NADPH oxidation, the electrons sequentially (one by one) travel along a chain of [4Fe4S]-clusters to reduce the FMN in StnB. The distinct iron-sulphur cluster environment surrounding the FMN cofactor in StnB is responsible for the electron bifurcating reaction of the complex. The FMN transfers the electrons to the clusters B2 and A1. The reduction of the B2 and C1 clusters triggers the binding of NAD+ at FMN. Another round of NADPH oxidation transfers two electrons to the binding site of the nucleotide, where FMN performs a hydride transfer to produce NADH. The reduction and dissociation of NADH triggers local rearrangements that lead to the outward "open" movement of StnA[17]. This allows electron transfer from B2 and A1 to the clusters B3-B4 in the C-terminal domain of StnB, which is the putative binding site for Fd reduction. The StnA subunit transits to a "closed" state upon the oxidation of cluster A1, preventing the backflow of electrons from the C-ter of StnB to FMN. The resulting conformational dynamics create a redox-driven kinetic gate, which ultimately enables the reduction of Fd. The conformational changes in StnAB are thermodynamically driven by the reduction and dissociation of the nicotinamide[17].

involved in stabilizing a semiquinonic state instead of stabilizing a fully reduced flavohydroquinone (Fig. 4e). The FMN coordination is different from other bifurcating complexes that have an arginine or lysine residue in the vicinity of the isoalloxazine ring to stabilize the fully reduced form of flavin (Supplementary Fig. 10). Further when comparing the two states of the structures, we were able to unveil dynamics associated with D186 in the presence and absence of bound NAD+ (Fig. 4e). Interestingly, this peculiar feature is not limited to the FMN in StnB, but has also been observed in the HydB protein and NADH:quinone oxidoreductase subunit 1 (Nqo1), where in Nqo1, glutamate (E97) is involved in the protonation of NAD+ instead of D186[17,25] (Supplementary Fig. 10). Along with D186, we also noticed subtle changes associated with E184 upon NAD+ binding to the FMN (Fig. 4e). Overall, we expect FMN to go through a transient one-electron reduced flavo-semiquinone state, which has also been proposed for HydABC[17,26]. This allows the FMN to transfer electrons sequentially to [2Fe2S]-clusters B2 and A1, which are located around 15 Å edge-to-edge distances from the FMN in the closed state (StnA-in) protomer (Fig. 4b). Previous studies with HydABC complexes from different organisms suggest that the thioredoxin-like C-terminal domain of HydC (or StnA) could move out in an "open conformation", allowing favourable electron transfer to the B3 and B4 clusters coordinated by the C-terminal domain of StnB, which is the putative site for the Fd reduction[17]. The conformational change in StnA is essential as it would prevent the backflow of electrons in a kinetically gated mechanism. To address whether this kinetic gating by StnA plays a functional role in the StnABC complex, we deleted the [2Fe2S]-cluster (StnABC_ΔA1) (Supplementary Fig. 8a, b). This variant indeed lacked two mol of iron in the complex compared to the WT, and in this variant, neither

the simultaneous reduction of NAD+ and Fd_ox with NADPH nor the NADH- and Fd_red-dependent reduction of NADP+ could be observed (Fig. 4d). Although the physiological activities were not catalysed, MV_ox was reduced with NADPH with an activity similar to the WT (89.72 U/mg) (Fig. 4d, Supplementary Table 1), which was expected since the subunit StnC harbours the NADPH:MV_ox oxidoreductase activity of the Stn complex[12]. Moreover, the mutational and biochemical analysis of StnABC_ΔA1 provides experimental evidence that the [2Fe2S]-cluster in StnA is important for shuttling the electrons to the C-terminal domain of StnB for the endergonic reduction of Fd. We can thus conclude that the FMN-containing StnAB module is used for bifurcation, similar to a proposed kinetic gated mechanism described for the electron bifurcating hydrogenase HydABC (Fig. 5, see Discussion)[17].

## Discussion

Nature has come up with two separate machines to catalyse the endergonic reduction of NADP+ with NADH to the exergonic reduction of NADP+ with reduced ferredoxin. The Nfn and Stn-type electron-bifurcating transhydrogenases are both widely distributed among bacteria and archaea[8,12,20,27]. Although they catalyse the same reaction, they are extremely different in structure and complexity and use fundamentally different mechanisms for electron bifurcation.

NfnAB has a comparatively simple structure to StnABC, it consists of only two subunits NfnA and NfnB. The FAD molecule of the NfnB (b-FAD) resides at the site of electron bifurcation, and other cofactors, including a [2Fe2S]-cluster, two [4Fe4S]-clusters, and the a-FAD molecule on the NfnA subunit, contribute to electron transfer[11,22]. Further, the short-lived anionic flavin semiquinone (ASQ) state of b-FAD is essential for electron bifurcation[21]. Nfn homologs are widely

distributed among bacteria and archaea, and probably eukaryotes[27]. Due to its simplicity in terms of the number of cofactors and clusters to work with, Nfn is the best understood bifurcating enzyme.

Stn, on the other hand, was discovered rather recently and has a different architecture. It consists of three subunits, one (StnC) is partly similar to NfnB and the others (StnAB) are similar to HydBC; there is no NfnA homologue in Stn and, conversely, there is no HydBC homologue in Nfn. Strikingly, StnC lost the ability to bifurcate electrons. Our cryo-EM structure and functional dissection of the electron transfer pathway of the StnABC complex revealed that it resembles an amalgamation of two different bifurcating types of machinery. The N-terminus of StnC has a GltD-nucleotide binding fold with very high similarity to the NfnB protein of the Nfn complex. It binds with an insertion of a NuoG domain to the electron bifurcating StnAB subcomplex. The StnAB subcomplex belongs to a widespread class of bifurcation modules related to the HydBC proteins of electron-bifurcating hydrogenase from anaerobic bacteria[12,17]. The described mechanism differs drastically from conventional FBEB enzymes[11,22,28–31] and is rather analogous to the bifurcation reaction that was initially described as a part of the Q-cycle in complex III (cytochrome bc1 reductase) within respiratory chains[32,33].

Our structure allows us to speculate how the GltD domain on one hand has evolved an FAD-binding pocket that directly allows for electron bifurcation in NfnAB, while it needs to associate with an additional bifurcation module in the Stn complex. At first glance, the binding site of the FAD cofactor is highly similar in StnC and NfnB. However, bifurcation in NfnB is made possible by supporting the formation of an anion in FAD by inhibiting N5 protonation by an interaction to R201[22]. This allows generating an ASQ that is essential for generating a low-potential electron for reducing Fd. Our structural analyses demonstrate that the binding pocket in StnC is slightly remodelled in a way that R239 does not form a hydrogen bond with N5. As a consequence, this would disfavour bifurcation but allows protonation of FAD⁻ and promotes the formation of an NSQ. Therefore, upon receiving two electrons from NADPH, the FAD forms an HQ (two electrons reduced state) and sequentially releases these two electrons via nearby clusters by fluctuating in two population states, which are the HQ/NSQ and NSQ/OQ (fully oxidised quinone state). The low activity of R239 mutants indicates two possible scenarios: (1) either the absence of an arginine residue destabilizes the binding of the FAD or the formation of NSQ, and/or (2) it tunes the redox potential in a way that disfavours electron transfer[23].

The second, obvious change is the [4Fe4S]-cluster proximal to FAD. In NfnB it has been shown that this cluster is coordinated by a glutamate residue. It has been suggested that the low potential of the proximal [4Fe4S]-cluster stems from coordination with a negatively charged glutamate residue that might allow for favourable electron transfer while maintaining the highly reducing nature of the [4Fe4S]-cluster[11]. In StnC we now observe in our structure that either a lysine or a cysteine residue coordinates the proximal [4Fe4S]-cluster in the StnABC_{S1} or StnABC_{S2} state, respectively. It is conceivable that the lysine coordination might modulate the redox potential in a way that favours electron transfer to the subsequent clusters. The coordination of a [4Fe4S]-cluster by a lysine residue observed here might have a similar effect on regulating the redox potential of iron sulphur clusters coordinated by an arginine[34,35]. Supporting that notion, the replacement of lysine against cysteine or alanine abolished any activity while introducing arginine retained at least half of the physiological activity. This is a strong indication that a residue that can be deprotonated and is able to provide a long electron pair to coordinate an iron ion in C7 is essential in this position. In line with this notion and our data, the 3.5 Å long density between C114 and one of the acid-labile sulphurs of C7 found in the StnABC_{S2} state may represent a hydrogen species, e.g., a proton abstracted from K170 or Y117. Something similar has been discussed in [FeFe]-hydrogenase where the protonation of an [4Fe4S]-

cluster shifts the catalytic redox potential by approximately +50 mV[36]. Alternatively, the observed density could stem from a sulphur atom of C7 dislocated by 3.5 Å that would produce a [4Fe3S]-cluster with a less negative redox potential[37]. Organometallic cofactors with mobile sulphur species have been reported for [NiFe]-hydrogenase[38], V-nitrogenase[39], and to a certain extent in CO dehydrogenase II[40]. However, with the current resolution, we are not able to distinguish between a hydrogen or sulphur species. In future work, we will investigate the redox-dependent changes in the iron sulphur cluster C7 by EPR and Raman spectroscopy. Our combined findings indicate a cooperative behaviour of K170 and C114, alternating the coordination of C7. This, in turn, will ultimately modulate the redox potential of C7 in the enzyme's oxidised and reduced condition, promoting rapid and unidirectional electron transfer to the FMN site in the associated StnAB subcomplex.

Based on our combined structural and biochemical data we can conclude that the StnAB module does not perform a classical FBEB mechanism[21,22]. It uses a bifurcation reaction guided by redox-driven conformational changes similar to the HydBC module from the HydABC complex[17] (Fig. 5). Consequently, during the first round of NADPH oxidation, two electrons are transferred stepwise from StnC to reduce the FMN in the StnB subunit. The FMN then facilitates electron transfer to the B2 and A1 [2Fe2S]-clusters, undergoing a one-electron reduction to form a neutral flavo-semiquinone state. The reduction of the B2/A1 clusters enhances the binding affinity for NAD⁺, as previously demonstrated[17]. In the second round of NADPH oxidation, a two-electron reduced state of the FMN is formed (FMNH⁻), which then performs a hydride transfer to NAD⁺ to generate NADH. Notably, in the stable reduced state of NADH/B2(red)/A1(red) observed in the StnABC_{S2} state, electron transfer to reduce Fd is hindered due to the long distance between the B2/A1 and B3/B4 clusters. We propose that the dissociation of reduced NADH triggers an outward conformational swinging motion of StnA, closing the gap between the A1 and B3 clusters and allowing efficient electron transfer (Fig. 5). Similarly, the electron from B2 is transferred to the B3/B4 cluster via the A1 cluster, ultimately leading to the reduction of two Fd molecules (Fig. 5). Subsequently, StnA reverts to its original ground state which increases the distance between the A1 and B3 clusters of StnA/StnB, thus, preventing the backflow of electrons from B3-B4 to the flavin and supporting the stepwise reduction of Fd. Our mutational analysis of the StnABC_ΔA1 cluster validates this proposed mechanism, as it results in the complete loss of bifurcation activity (Fig. 4d).

Overall, our study demonstrates, that the hydrogenase subunit of HydABC (HydA) can be functionally replaced with another electron input module, such as an NADPH oxidising subunit (StnC), and that NADPH oxidation can be reversibly coupled to the reduction of NAD⁺ and Fd in the bifurcation module (StnAB/HydBC). This suggests that the bifurcating module can be linked to a broader array of metabolic processes, as suggested by a phylogenomic analysis[20], and that there may exist an additional diversity within the HydBC family that could be identified in future research. Furthermore, our study highlights how nature has come up with two solutions for electron bifurcation in enzymes containing the GltD domain. The amalgamation of structures also illustrates the principle of modular evolution in anaerobic metabolism through domain insertion and association with functionally differentiated redox modules to adapt to a variety of different and extreme environmental conditions. A theme that recurs but has certainly been taken to extremes in the Stn transhydrogenase complex.

## Methods

### Growth of *S. ovata* and purification of the Stn complex
Purification of the Stn complex was performed as described before[12]. *S. ovata* was grown at 30 °C under strictly anoxic conditions in 20 l of medium DSM 311. $Na_2S \times 9\ H_2O$ were omitted and

0.6 g/l cysteine-HCl × H$_2$O was used as reducing agent, instead. Instead of glycine-betaine, fructose (20 mM) was added as carbon and electron source. Cells were harvested at late exponential growth phase (OD$_{600}$ = 2) and washed twice in buffer A (50 mM Tris/HCl, pH 7.5; 20% glycerol, 20 mM MgSO$_4$, 5 µM FAD, 2 mM DTE, 4 µM resazurin). Cells were resuspended in buffer A and 0.5 mM PMSF and 0.1 mg/ml Dnase I were added, before disruption of the cells using a French Pressure Cell (110 MPa). Cell debris were removed by centrifugation (24,000 × g for 45 min). Cytoplasmic- and membrane fraction were separated by ultracentrifugation (210,000 × g for 45 min) and the cytoplasmic fraction was used for purification of the Stn complex using FPLC. First, the cytoplasmic protein was applied to anionic exchange chromatography on Q Sepharose®. Column-bound protein was eluted by applying a linear gradient from 0 to 400 mM NaCl. Fractions with NADPH:MV$_{ox}$ oxidoreductase activity were collected and (NH$_4$)$_2$SO$_4$ was added to the protein solution to a concentration of 2 M. Precipitated protein was removed by centrifugation (24,000 × g for 30 min) and the soluble protein was applied to hydrophobic interaction chromatography on Phenyl Sepharose®. The column was washed by decreasing the concentration of (NH$_4$)$_2$SO$_4$ to 1.4 M. Column-bound protein eluted applying a liner gradient from 1.4 M to 0 M (NH$_4$)$_2$SO$_4$. NADPH:MV$_{ox}$ oxidoreductase activity was measured, active fractions were collected and applied to size exclusion chromatography on "Superdex™ 200 Increase 10/300" (equilibrated with buffer B = buffer A + 250 mM NaCl). Again, active fractions were collected and applied to group-specific interaction chromatography on Blue-Sepharose® equilibrated with buffer B. Protein was eluted by increasing the NaCl concentration from 250 mM to 1 M. Elution resulted in a single protein peak, containing the purified Stn complex.

## Crosslinking of the Stn complex

All steps of crosslinking were performed inside an anaerobic chamber. To crosslink the Stn complex the protein-containing buffer was changed to HEPES buffer 1 (50 mM HEPES, 150 mM NaCl, 20% glycerol, pH 7.5) and the concentration of the protein was adjusted to 1 mg/ml. Bis(sulfosuccinimidyl) suberate was added to a final concentration of 1 mM to start the crosslinking reaction. The mixture was incubated for 30 min at room temperature before Tris was added to a final concentration of 50 mM to quench the reaction. Unreacted crosslinker was removed by gel filtration on "Superdex™ 200 Increase 10/300" which was equilibrated with HEPES buffer 2 (50 mM HEPES, 150 mM NaCl, 5% glycerol, 10 µM FAD, pH 7.5). To detect the interaction of cofactors with the Stn, the complex was crosslinked in presence of NADPH (500 µM), NAD$^+$ (500 µM) and Fd$_{ox}$ (30 µM) in HEPES buffer 3 (=HEPES buffer 2 + 10 µM FMN).

## Construction of pET21a_StnABC-His

Genomic DNA of *S. ovata* was used as template in a PCR using the oligonucleotides 1 and 2 (Supplementary Table 4), whereas pET21a_StrepStnC[12] was used as template in a PCR with oligonucleotides 3 and 4 (Supplementary Table 4). The PCR-products were assembled by the method of Gibson to build the vector pET21a_StnABC. Sequencing of the plasmid revealed two mutations at positions 13 (A->C) and 15 (T->C) of the non-coding regions between StnB and StnC. Furthermore, the plasmid contained a silent mutation of the alanine encoding GCA-codon starting at position 3187 of StnC (GCA to GCC). A His-Tag encoding sequence was added to the C-terminus of StnC by amplification with oligonucleotides 5 and 6 and following blunt-end ligation. The resulting vector pET21a_StnABC-His was used as a template to introduce targeted mutations into the open reading frame of the Stn using the corresponding oligonucleotides (Supplementary Table 4). The introduction of targeted mutations (Supplementary Table 5) was verified by sequencing.

## Overproduction and purification of Stn variants

The Stn-variants were overproduced in *E. coli* BL21 (DE3) *ΔiscR* as described before[41], but cells were grown in a volume of 2 l and protein production was induced at a cell density of OD$_{600}$ = 1.2–1.5 by adding IPTG to a final concentration of 0.5 mM. Under anoxic conditions, cells were harvested by centrifugation (14,000 × g, 10 min, 4 °C) and washed once in oxygen-free buffer 1 (50 mM Tris/HCl, 20 mM MgSO$_4$, 20% glycerol, 300 mM NaCl, 0.5 mM DTE, 5 µM FAD, pH 7.5). Cells were disrupted using a French Pressure Cell (1 × 110 MPa). Cell debris and membranes were removed by ultracentrifugation (220,000 × g, 30 min, 4 °C). The resulting cytoplasm was incubated with Ni-Sepharose for 30 min and the variants were purified according to the manufacturer's protocol using buffer 2 (buffer 1 + 20 mM imidazole) for equilibration and washing steps and buffer 3 (buffer 1 + 250 mM imidazole) for elution. The eluate was further purified by gel filtration on "Superdex™ 200 Increase 10/300", which was equilibrated with buffer 4 (buffer 1 without NaCl).

## Enzymatic measurements

All enzyme assays were performed in 1.8 ml cuvettes (Glasgerätebau Ochs, Bovenden-Lenglern, Germany) sealed by rubber stoppers under an N$_2$-atmosphere. NADPH-dependent simultaneous reduction of ferredoxin (Fd$_{(ox)}$) and NAD$^+$, as well as Fd$_{red}$- and NADH-dependent reduction of NADP$^+$ were performed at 50 °C in buffer A (50 mM Glycine-Glycine, 10 mM NaCl, 20 mM MgSO$_4$, 2 mM DTE, 4 µM Resazurin, pH 8) as described before[12]. When simultaneous reduction of ferredoxin and NAD$^+$ with NADPH was measured, the level of NADPH was kept constant using an NADP$^+$ reducing system consisting of 0.25 mM NADP$^+$, 1 unit glucose-6-phosphate dehydrogenase and 20 mM glucose-6-phosphate. After a constant level of NADPH was reached, the protein fraction was added and 30 µM ferredoxin and 0.5 mM NAD$^+$ were added to start the reaction. Oxidation and reduction of NAD(H)/NADP(H) were monitored at 340 nm (ε = 6.3 mM$^{-1}$ cm$^{-1}$) whereas the reduction of ferredoxin (purified from *Clostridium pasteurianum*[42]) was monitored at 430 nm (ε = 13.1 mM$^{-1}$ cm$^{-1}$). For NADP$^+$ reduction catalysed by the Stn complex, 30 µM ferredoxin were pre-reduced with CODH using a CO-atmosphere. 0.25 mM NADH and 0.5 mM NADP$^+$ was added and the change of absorbance at 340 nm was followed to determine NADP$^+$ reduction. Since NADH oxidation leads to a change in absorbance at 340 nm as well, the rate of NADP$^+$ reduction has been corrected by the NADH oxidation rate (assuming a ratio of 1 NADH oxidised per 2 NADP$^+$ reduced). NADPH-dependent reduction of methyl viologen was performed at room temperature (23 °C) in buffer B (100 mM Tris/HCl, 2 mM DTE, 4 µM resazurin, pH 7.5) and monitored at 604 nm (ε = 13.9 mM$^{-1}$ cm$^{-1}$) as described before[12]. 5 mM methyl viologen and 0.5 mM NADPH were used as substrates. One Unit is defined as the transfer of 2 µmol electrons/min. WinASPECT 2.5.0.0 software was used for analysing the photometric data.

## Analytical methods

The protein concentration was determined according to Bradford[43] using Bovine serum albumin (BSA) as a standard. Proteins were separated in 12% polyacrylamide gels according to Laemmli[44] and stained with coomassie brilliant blue G250. The iron content was determined calorimetrically according to Fish[45]. The nature of the flavin was determined as previously described[46]. Size exclusion chromatography was performed[12] and analysed using UNICORN 5.31 software.

## Mass photometry measurements

The mass photometry was performed using OneMP mass photometer, and data was collected using AcquireMP. MP movies were recorded at 1 kHz, with exposure times varying between 0.6 and 0.9 ms, adjusted to maximize camera counts while avoiding saturation. The slides (70 × 26 mm$^2$) were cleaned using a solution of 50% (v/v) isopropanol and Mili-Q water. Just before the measurements, silicon gaskets were also

cleaned and fixed to clean glass slides, followed by calibrating the instrument. At the beginning, 18 μL of buffer (50 mM HEPES, 150 mM NaCl, 5% glycerol, 10 μM FAD, pH 7.5) was added to find the focus, which was followed by adding 2 μL of sample to the well, resulting in a 10-fold dilution for recording single events. The final concentration of the protein complex for mass photometry measurements was 25 nM. The data analysis was performed using DiscoverMP software.

### Cryo-EM data acquisition, processing, and model building

For cryo-EM data acquisition of StnABC, all handlings were carried out in an anaerobic glove box with a gas composition of 95% $N_2$ and 5% $H_2$ (vitrobot was placed inside this anaerobic tent). The natively purified StnABC complex was stabilized by mild cross-linking before grid preparation inside an anaerobic tent. To obtain the structure in purified state (which we call as $StnABC_{S1}$ state), 3.5 μl of 1 mg/ml of mildly crosslinked protein complex without any added cofactor (FMN) and substrates (NADPH, NAD$^+$, and Fd) was rapidly applied to glow-discharged Quantifoil grids, blotted with force 4 for 3.5 s, and vitrified by directly plunging in liquid ethane (cooled by liquid nitrogen) using Vitrobot Mark III (Thermo Fisher) at 100% humidity and 4 °C. The automated data acquisition was performed using SerialEM software on an FEI Titan Krios transmission electron microscope operated at 300 keV, equipped with a K3 Summit direct electron detector (Gatan). Movie frames were recorded at a nominal magnification of ×22,500 with an electron dose of 55 e- per Å$^2$ spread over 30 frames at a calibrated pixel size of 1.09 Å. A total of 3008 images were acquired using SerialEM[47]. The structure of $StnABC_{S2}$ state was determined by subjecting the purified StnABC complex to mild crosslinking in presence of exogenous cofactor and substrates (10 μM FMN, 500 μM NAD, 30 μM Fd, 500 μM NADPH). 4 μl of 1 mg/ml of this protein was applied to glow discharged Quantifoil grids, blotted for 4 s with force 5, and flash-frozen into liquid ethane using Vitrobot Mark IV (Thermo Fisher) at 100% humidity and 4 °C. The data collection was performed using SerialEM on a Jeol CryoArm200 microscope operated at 200 kV and equipped with a K2 direct electron detector. Frames were acquired at a magnification of ×60,000 with a dose rate of 50 e$^-$ per Å$^2$ which were fractionated into 50 frames at a calibrated pixel size of 0.84 Å. A total of 2745 images were acquired.

Both datasets were processed in CryoSparc[48]. For both datasets, the frames were gain-normalized, aligned, dose-weighted using Patch Motion correction, and followed by CTF estimation. Manual picking of particles, followed by multiple rounds of topaz[49] which resulted in "good" reference-free 2D classes that were further used for the processing. For StnABC state 1 ($StnABC_{S1}$), 326,306 particles were used for creating three ab-initio classes, out of which only 272,183 particles exhibiting a nice ab-initio model were used for a single round of heterogeneous refinement. The class exhibiting desired features with 116,573 particles was then used for the final round of NU refinement. The refinement yielded 3D reconstructed maps for $StnABC_{S1}$ state with a resolution of 3.23 Å with D2 symmetry. For StnABC state 2 ($StnABC_{S2}$), after 2D classification, 96,166 particles were cleaned by generating three ab-initio classes, followed by a heterogeneous refinement. This resulted in 54,596 particles which were further refined to a resolution of 3.0 Å with D2 symmetry imposed.

Initial models of StnA, StnB, and StnC from *S. ovata* were generated separately from their protein sequences using alphaFold[15], and thereupon fitted as rigid bodies into the density using UCSF Chimera[50]. The model was manually rebuilt using Coot[51] with cofactors placed manually and refined by using their respective CIF files. The final model was subjected to real-space refinements in PHENIX[52]. Illustrations of the models were prepared using UCSF Chimera, UCSF ChimeraX[53] and PyMOL[54].

### Reporting summary

Further information on research design is available in the Nature Portfolio Reporting Summary linked to this article.

## Data availability

The data that support this study are available from the corresponding authors upon request. Cryo-EM maps are available in the Electron Microscopy Data Bank; StnABC state 2 ($StnABC_{S2}$) (16878) and StnABC state 1 ($StnABC_{S1}$) (16879). The atomic models of StnABC are available in the Protein Data Bank: StnABC state 2 ($StnABC_{S2}$) 8OH5 and StnABC state 1 ($StnABC_{S1}$) 8OH9. Structural and sequence data used for comparison with StnABC subunits are available in the Protein Data Bank: 6Q8W, NADH:quinone oxidoreductase from *Thermus thermophilus*; 5JCA, NADP(H) bound NADH-dependent Ferredoxin:NADP oxidoreductase (NfnAB) from *Pyrococcus furiosus*; 7NZ1, NADH:ubiquinone oxidoreductase from *Escherichia coli*; 8A6T, electron bifurcating Fe-Fe hydrogenase HydABC complex from *Thermoanaerobacter kivui*; 1H0H, Tungsten containing Formate Dehydrogenase from *Desulfovibrio gigas*; 3FX2, crystal structure of Flavodoxin. Strains and plasmids generated in this study are available from the corresponding authors upon request. Source data are provided with this article. Source data are provided with this paper.

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

## Acknowledgements

We thank Dr. Stefan Bohn from Helmholtz Zentrum Munich and the cryo-EM facility at University Marburg for microscope access and data acquisition. We are grateful to Dr. Clinton Gabel, Dr. Sven T. Stripp, and Dr. Joseph Braymer for useful discussions and critical reading of the manuscript. We also thank Dr. Georg Hochberg and Stefano Lometto for helping and collecting the mass photometry data. We acknowledge the contribution of the Core Facility 'Protein Biochemistry and Spectroscopy'

of the Philipps University of Marburg. J.M.S. acknowledges the DFG for an Emmy Noether grant (SCHU 3364/1-1) and the European Union's Horizon 2020 research and innovation programme (Two-CO2-One, grant agreement no. 101075992). This work received funding from the German Research Foundation (DFG) and the European Research Council under the European Union's Horizon 2020 research to V.M.

## Author contributions

F.K. isolated and biochemically characterized the protein and performed site-directed mutagenesis experiments; J.R. determined the Fe content; S.F. and A.K. prepared cryo-EM grids. A.K. collected data, processed cryo-EM data, and built models; A.K., V.M. and J.M.S. analysed and interpreted the cryo-EM models; A.K., F.K., V.M. and J.M.S. wrote the manuscript with input from all authors.

## Funding

## Competing interests

The authors declare no competing interests.
