## [Peer Review File · Nature Communications]

Molecular architecture and electron transfer pathway of the Stn family transhydrogenaseREVIEWER COMMENTS

Reviewer #1 (Remarks to the Author):

Key findings are the domain-scale homologies displayed by this new structure, and the way in which the structure recombines domains familiar from other enzyme complexes.

The work will advance our understanding of enzyme evolution, and mechanisms for balancing pyridine nucleotide pools.

Numerous assertions are made without reference to data. Data may well exist but this paper often does not clearly explain how the observations/data are responsible for many of the statements made. Similarly, statements and interpretations are given that are not supported by the resolution in hand. Given the resolutions around 3 Å, that already incorporate assumptions from models, only displacements on the order of 0.5 Å and larger are amenable to confident interpretation.

Although mutations were made to produce variants, and their activities are reported, the activities are difficult to interpret in the absence of any information whatsoever regarding the stoichiometry with which Fe and flavins are retained. This information should be provided for each of the samples.

The writing does not do the science justice at all. This will discourage many readers and rob the authors of some of the credit they deserve.

Detail provided is insufficient to fully understand the figures, and certainly insufficient to permit reproduction of the work.

Specific comments follow:

Throughout, the paper needs the attentions of a fluent English speaker to correct usage, plural/singular incompatibilities, verb tenses and wording. This problem is compounded by very long sentences and convoluted clauses. These conspire to degrade the clarity of the narrative.

The abstract needs to be completely rewritten, and there is ample room for improvement in all other sections as well. This effort will enable readers to better appreciate the exciting science.

What is the Fe and flavin stoichiometry of each of the preparations? This is crucial to interpretations of the mutations and deletions.

What is the experimental evidence demonstrating which of FAD/FMN is the site of bifurcation?

Why do the authors think that FMN switches between exergonic and endergonic electron transfer modes? The two transfers involved in bifurcation must be tightly coupled, which is to say neither occurs alone. Thus there is a single reaction involving both transfers.

There is considerable confusion regarding whether bifurcation requires a stable semiquinone or a high-energy semiquinone. These are mutually exclusive so please determine which applies and explain how it can participate in bifurcation. For an exceptionally clear treatise, please see Nitschke and Russel, reference 10.

Please make a clear consistent distinction between bound cofactors (prosthetic groups) and substrates/products. The cofactors (prosthetic groups) are those that remain bound to the enzyme: FeS clusters, Zn, FMN and FAD, whereas NAD(H) and NADP(H) have the status of substrates and products in this reaction. Even Fd is technically a substrate/product in this case.

Please list the complement of cofactors (=prosthetic groups) present in the 'oxidized' samples, for comparison with those present in the reduced samples. In its current form, the manuscript says that cofactors are absent from the oxidized protein. If it is genuinely apoprotein then it is no wonder that there is increased disorder in some places and displacement. Hopefully only NAD(H) and NADP(H) are absent? Moreover if simply not adding them to the buffer results in their absence from the structure, this will confirm that they are relatively weakly bound and thus that they do not have the status of cofactors.

Maybe there is an issue in translating terminology to English, but we understand that a cofactor assists in catalysis. NAD(H) and NADP(H) are certainly recycled in cells, but for any given reaction, each is either a substrate or a product. (Exceptions include cases where a pyridine nucleotide is an allosteric effector.)

A claim is made that because a portion of the protein is disordered, the clusters it contains cannot be important for catalytic electron transfer. Why? Could the disorder represent static disorder stemming from partial occupancy?

Please define all abbreviations at the time of first use.

How was it determined that the Zn is 'strongly' bound? bond lengths will not be precise enough to arbitrate at this resolution.

What are the oxidation states of the bound cofactors (prosthetic groups) of the 'OX sample' and how do they compare with the same in the reduced sample? Please at minimum provide optical spectra of each and a difference spectrum (scale the two parent spectra to the same enzyme concentration before subtracting). The 'Methods' says that protein complex without any FMN, NAD⁺, NADPH or Fd was used for the 'Oxidized' sample. In what respect is it oxidized? Was it exposed to air?

Given that protein was crosslinked prior to deposition on grids, we do not expect to see extensive conformational sampling. Correct? The text nevertheless refers to 'dynamics'. Would it be better described as disorder? That would lower the resolution of the region in question but represent static disorder rather than dynamics. It could mean that the local structure is not strongly constrained to a unique conformation, rather than that individual regions of protein move back and forth on an ongoing basis.

It is unclear what evidence forms the basis for the proposed sequence of events in the mechanism. Moreover the mechanism needs a much clearer description please.

Considering that the structural resolution is 3 Å, and that is itself substantially based on modeling, it is premature to discuss subtle orientational details, such as the orientation of the terminal guanidinium of an Arg.

Please specify what operations were conducted under inert atmosphere. Readers need to be told this, as many will not be familiar with air-sensitive enzymes.

For the Bradford assay, what protein was used for the standard curve?

Caption of Figure 5: the text implies that both electrons from NADPH travel down a chain of Fe₄S₄ clusters, but FeS clusters are normally carriers of single electrons. Please clarify what is being proposed.

Figure captions in general lack needed detail.

Legend in Figure S9, there is a typo in 'So StnB Redcued' (should read 'Reduced').

Reviewer #2 (Remarks to the Author):

What are the noteworthy results?

CryoEM structures of StnABC from *Sporomusa ovata* along with mutagenesis and activity assays to probe mechanistic features. See attached review for details.

Will the work be of significance to the field and related fields?

Yes. See attached review for details.

How does it compare to the established literature? If the work is not original, please provide relevant references.

The work is highly original. See attached review for details.

Does the work support the conclusions and claims, or is additional evidence needed?

Overall, yes, but I recommend some additional discussion to clarify some points. See attached review for details.

Are there any flaws in the data analysis, interpretation and conclusions? Do these prohibit publication or require revision?

There may be some minor flaws. See attached review for details.

Is the methodology sound? Does the work meet the expected standards in your field?

For the most part, but there are some minor questions on this that I would like to see addressed. See attached review for details.

Is there enough detail provided in the methods for the work to be reproduced?

Yes.

The manuscript “*Molecular architecture and electron transfer pathways of the Stn family transhydrogenase*” by Kumar, Kremp and coworkers reports cryoEM structures of the electron-bifurcating transhydrogenase from *Sporomusa ovata* (StnABC) in an “oxidised” and a “reduced” state. The authors find that the enzyme forms a dodecameric structure comprising four functionally independent StnABC heterotrimers. The structure and arrangement of cofactors is similar to other related electron-bifurcating enzymes, and a similar mechanism of electron-bifurcation at the FMN cofactor in the StnB subunit is proposed. Interestingly, changes in the FAD and proximal [4Fe-4S] cluster environment in the StnC subunit are proposed to be responsible for changing this flavin from an electron-bifurcating form in the related Nfn complex to a non-electron bifurcating Flavin in StnABC. The authors were also able to produce the enzyme heterologously in *E. coli* and used this system for targeted mutagenesis. Overall, I find this to be an interesting and well executed study and I can recommend publication after consideration of some minor points and slight clarification of the text.

The lack of FMN in the “oxidised” enzymes has not been adequately addressed. At the top of Page 5 it is stated that:

“For reducing the enzyme, a mixture of cofactors was added to initiate the reaction, followed by immediately vitrifying the grids (see methods). The structure under oxidizing conditions was obtained by adding no cofactors.”

In the methods, this is specified to mean that FMN, NAD⁺, Fd and NADPH were added or not. Table 2 implies that the oxidised enzyme lacks FMN entirely and at the top of Page 6 it is stated that:

“The structure solved under oxidizing conditions only contained FAD in the StnC subunit, while under reducing conditions FMN-NAD⁺ was also found to be present in the StnB subunit.”

If the oxidised enzyme lacks FMN then how is it able to perform catalysis? The activity assays reported on Page 4 do not appear to have been carried out in the presence of exogenous FMN.

More generally, it is not clear why the oxidised enzyme lacks FMN, why the authors did not deem it sensible to attempt reconstitution of FMN for their oxidised structure, and whether the differences between oxidised and reduced enzymes simply reflect the lack of FMN. In the methods it is explained that 5 μM FAD was included in the protein purification buffers, presumably to counteract loss of FAD. Why was FMN not also included? On Page 9 it is stated that:

“The comparison of StnABC structures in the oxidized and reduced state revealed no major differences, however, we could see the loop (residues 182–191) of StnB undergoing a significant conformational change upon NAD⁺ binding (Fig. 4e).”

How do the authors know that this change is not simply due to binding of FMN? Overall, I think the authors should be more transparent about the lack of FMN in the oxidised structure and rationalise why they observe enzyme activity for this preparation e.g. with reference to the higher catalytic rates observed in the presence of exogenous FMN reported in reference 12.

On the topic of catalysis, I was struck by the observation that NADPH-dependent ferredoxin reduction (1.59 U/mg) and NAD⁺ reduction (2.80 U/mg) do not have the same rates. The authors should clarify the reason for this. It cannot be due to reduction of only one of the two iron-sulphur clusters in ferredoxin (indicated in Figure 5), as the rates are calculated based on “*One Unit is defined as the transfer of 2 μmol electrons/min.*” Thus, it would appear that 36% of electrons go to ferredoxin and 64% of electrons go to NAD⁺. Is there some ferredoxin:NAD⁺ oxidoreductase activity? Could the authors clarify this in the current manuscript with reference to the activity measurements reported in reference 12? From a brief look at reference 12, it appears that there is both a ferredoxin-independent NADPH:NAD⁺ oxidoreductase rate and a Fd_{red}:NAD⁺ oxidoreductase rate, which presumably distort the stoichiometry. Related to this, could the authors explain how they monitor NADH and NADPH independently even though they have the same absorbance spectrum. Presumably nucleotide-specific regeneration systems

are used and assumed to keep the absorbance or either NADPH or NADH constant as reported in reference 12?

I was also struck by the lack of the plant-type ferredoxin [2Fe-2S] cluster in the N-terminal NuoG-like domain of StnC (equivalent to N1b in NuoG/Nqo3 and A4 in HydABC). Could the authors add a brief comment about this? The cluster binding motif appears to be present so it is surprising that there is no evidence for this cluster in the structure.

Minor points:

1. Abstract: “*sustain*” > “*sustain life*”
2. Page 2: “*adenosinetriphosphate*” > “*adenosine triphosphate*”
3. Page 3: “...*ratio of NAD⁺/NADH is 30/1...*” – ratios normally given with “:”
4. Page 3: “*Contrary...*” > “*On the contrary...*” or “*In contrast...*”
5. Page 3: “...*anaerobes do neither have...*” > “...*anaerobes neither have...*”
6. Page 3: “...*for long time.*” > “...*for a long time.*”
7. Page 3: “...*NADPH to NAD drives...*” > “...*NADPH to NAD⁺ drives...*”
8. Page 3: “*a semiquinone anion is stabilized resulting in two very different redox potentials (“crossed potentials”) that allow one-electron transfers to acceptors with high and low potentials¹⁰.*” This should be clarified. It is possibly simply a typo but I think “*stabilized*” should be replaced with “*destabilized*”. In my understanding of reference 10 “crossed potentials” indicates that the redox potential of the semi-reduced/fully-reduced pair is more positive than the redox potential of the oxidised/semi-reduced pair, meaning that the semi-reduced state is destabilized. Additionally, on Page 9 it is stated that “*The FMN coordination is different from other bifurcating complexes which have an arginine or lysine to stabilize an ASQ for using the concept of cross potential where a flavin stabilises the ASQ using an Arg/Lys residue to release two electron on spatially separated pathways with very different potentials (Fig. S9).*” Again, I don’t think that stabilization of the ASQ leads to crossed potentials.
9. Page 3: “*In addition to the bifurcating flavin, a second flavin is required to collect the single electrons before a hydride can be transferred to NAD⁺ or NADP⁺.*” While this is indeed the case for Nfn and a number of other FBEB enzymes, it is not the case for Sfn or HydABC. Perhaps the word “*usually*” should be inserted between “*is*” and “*required*”?
10. Page 3: “...*a comparable simple...*” > “...*a comparably simple...*”
11. Page 3: “...*NfnAB.*” > “...*NfnA and B.*”
12. Page 4: “...*low potential FADH^o radical...*” > “...*low potential FADH[•] radical...*”
13. Page 4: “*Interesting, Stn has...*” > “*Interestingly, Stn has...*”
14. Page 5: “*All cryoEM grids were prepared in a strictly anoxic chamber with a gas composition of 95% N₂ and 5% H₂, i.e., under catalytic turnover conditions.*” Why do these conditions imply catalytic turnover? Aren’t nucleotides and ferredoxin are required for this?
15. Page 5: “...*using Alphafold...*” – here it would be appropriate to cite reference 44.
16. Page 5: “*The tetramer comprises of four monomeric StnABC heterotrimeric complexes...*” – 1. Delete “*of*”, 2. What is a monomeric heterotrimeric complex? Surely by definition a heterotrimer is trimeric? Perhaps rephrase this for clarity.
17. Page 5: “...*six [4Fe-4S] cluster...*” > “...*six [4Fe-4S] clusters...*”
18. Page 6: “...*degenerate [Fe]-only hydrogenase...*” What is meant by degenerate? Also, the term [FeFe]-hydrogenase is used on Page 4. If the authors wish to refer to the same class of enzymes then I recommend using the same nomenclature.
19. Page 7: “...*simultaneous reduction of NAD⁺ and Fd_{ox} with NADP⁺...*” Presumably NADPH was meant?
20. Page 7: “...*the metals the [4Fe-4S] cluster...*” > “...*the metals of the [4Fe-4S] cluster...*”
21. Page 8: “...*electron-bifurcating hydrogenases from Thermoanaerobacter kivui...*” Are there more than one?

22. Page 8: "...which represents the binding site for Fd reduction..." Was this proven? I thought it was just a hypothesis.
23. On Page 9 it is stated that: "Overall, we expect FMN to go through a transient one-electron reduced flavo-semiquinone state, which has also been reported for HydABC²¹." Reference 21 does not report any evidence for a flavo-semiquinone state. This reference proposes the existence of one. In a supplementary figure (S11) from Chondgar et al., (2020) *JBIC* some evidence for an FMN radical can be observed in EPR spectra of HydABC in the presence of NAD⁺.
24. Page 11: "The described mechanism differs drastically from conventional FBEB enzymes^{11,19,24-27} and is rather analogous to the bifurcation reaction that was initially described as a part of the Q-cycle in complex III (cytochrome bc1 reductase) within respiratory chains^{28,29}." In what way is the electron-bifurcation reaction of StnABC analogous with the Q-cycle in complex III?
25. Page 11: "...the GltD domain of the Stn complex has evolved an FAD-binding pocket that allows electron bifurcation..." Is Nfn meant here?
26. Page 12: "...favours electrons transfer..." > "...favours electron transfer..."
27. Page 12: "...may represent a hydrogen species, e.g., a proton..." Is this realistically observed in cryoEM density at this resolution? Is the density attributed to other nearby protons distinguishable? I would rather favour the idea of a water molecule or change to the cluster geometry.
28. Page 12: "...in the enzymes oxidised and reduced condition..." > "...in the enzyme's oxidised and reduced condition..."
29. Page 12: "In summary, our study highlights how nature has come up with two solutions for electron bifurcation in the GltD domain." But I had understood that the GltD domain in StnC was not electron bifurcating? Please rephrase this sentence.
30. Figure 2 caption: "...one [2Fe2S] clusters..." > "...one [2Fe2S] cluster..."
31. Figure 2 caption: Ångström appears to have an Å (with an accent) rather than an A
32. Figure 3 caption: "...two Rossmann fold..." > "...two Rossmann folds..."
33. Figure 4f: a slightly unconventional nomenclature is used for the iron-sulphur clusters in Nqo1-3. I recommend using the nomenclature reported in reference 22, Figure 1B. Also, it isn't clear why this particular orientation of the cofactors in Nqo1-3 has been chosen as it doesn't align well with the related views from StnAB and HydBC. Furthermore, clusters have been included in the Nqo1-3 view that have related clusters in StnAB and HydBC
34. Figure 4 caption: "...nearby A1 and B2 cluster..." > "...nearby A1 and B2 clusters..."
35. Throughout: please harmonise the iron-sulphur cluster nomenclature. Sometimes [4Fe-4S] is used, sometimes [4Fe4S] and sometimes 4Fe4S. Same for [2Fe-2S]
36. Throughout: a mixture of english english (e.g. coloured, homologue) and american english (e.g. colored, homolog) is used. Please harmonise.

Reviewer #3 (Remarks to the Author):

Anuj Kumar et. al present the cryo-EM structure of Stn class transhydrogenase StnABC complex from *Sporomusa ovata* showing a tetramer of trimeric arrangement. The authors demonstrate its unique structure and function characteristics by necessary functional experiments and structure comparison with the known NfnAB and HydABC hydrogenase structures.

The text and figures are clear. The methods and results, including cryo-EM, are sound. I recommend this manuscript for publication in Nature Communications

I have a suggestion for the authors:

The relevant experiments were conducted under strictly controlled conditions (reduction or oxidation), and structural and functional analyses suggest that a single protomer is the minimal functional unit. These imply that the structure of different protomers should be uniform. However, it is always worth using symmetry expansion and/or local refinement (all available in cryoSPARC and Relion) to investigate the “personality” of protomers in an oligomerization complex to see whether the complex is indeed symmetrical or pseudo-symmetric.

Reviewer #4 (Remarks to the Author):

Short transhydrogenase (Stn) is a recently discovered flavin-based electron bifurcation (FBEB) enzyme that belongs to the HydABC superfamily. Kumar et al. reports the enzyme structure and a structural-based mutagenesis and activity assay. The work reveals the enzyme architecture, electron transport paths, and several unique features, leading the authors to propose a plausible electron bifurcation mechanism. The manuscript is well written, but we have a few concerns as detailed below. Overall, this work is a valuable contribution to the field.

Major concern:

1. The states of the samples need to be better explained. Both oxidized- and reduced-state samples were crosslinked and prepared in an anaerobic chamber. The difference was that the authors added additional cofactors to the reduced state. Why was the other sample without additional cofactors assigned oxidized state? Can you compare the structure of this sample with an air-oxidized sample?

2. The authors have solved the StnABC structure as a tetramer. But in Fig. S7 panel d, the WT StnABC was a monomer and dimer in the solution according to mass photometry measurements. Can you explain the observed monomer and dimer states? Is it possible that the mild crosslinking fixed the complex into an artificial state? If so, they should determine the structures of the uncross-linked monomer and dimer which are large enough for cryo-EM.

3. The EM density map in the core region of the complex looks decent. But the local resolution of several other regions, such as StnA, the N-terminal domain, and the C-terminal domain of StnB are difficult to assess due to the large map size, as these regions may not have reached 4 Å resolution, the upper color display bound used in the local resolution figures. The authors should revise these figures to better show these regions and with a color range higher than 4Å. The authors should also explicitly acknowledge that these regions do not allow atomic modeling but can fit the AlphaFold models.

4. The EM density of all cofactors should be presented (if not already shown in Fig. S4 or other figures). The authors should improve the modeling of NADP: the model does not fit the density well, and there are many severe clashes with the surrounding protein residues.

Minor:

1. Results section Purification of the Stn complex, please highlight Methyl Viologen when first mentioned with MV since the authors used the initials in the rest of the manuscript.

2. Fig. S7 panel c the SDS-PAGE is not discussed in the legend.

3. There are two D471 labels in panel C of Figure 3.

4. Discussion page 11, "Our study demonstrates, for the first time, that the hydrogenase domain of HydABC can be functionally replaced with another redox enzyme...". This is not the first structural evidence. Structural available [NiFe]-hydrogenase ABCSL and HydABC have very different enzyme subunits coupled to the bifurcation module.

Reviewer #1 (Remarks to the Author):

Key findings are the domain-scale homologies displayed by this new structure, and the way in which the structure recombines domains familiar from other enzyme complexes. The work will advance our understanding of enzyme evolution, and mechanisms for balancing pyridine nucleotide pools.

Numerous assertions are made without reference to data. Data may well exist but this paper often does not clearly explain how the observations/data are responsible for many of the statements made. Similarly, statements and interpretations are given that are not supported by the resolution in hand. Given the resolutions around 3 Å, that already incorporate assumptions from models, only displacements on the order of 0.5 Å and larger are amenable to confident interpretation.

Although mutations were made to produce variants, and their activities are reported, the activities are difficult to interpret in the absence of any information whatsoever regarding the stoichiometry with which Fe and flavins are retained. This information should be provided for each of the samples.

The writing does not do the science justice at all. This will discourage many readers and rob the authors of some of the credit they deserve. Detail provided is insufficient to fully understand the figures, and certainly insufficient to permit reproduction of the work.

Specific comments follow:

Throughout, the paper needs the attentions of a fluent English speaker to correct usage, plural/singular incompatibilities, verb tenses and wording. This problem is compounded by very long sentences and convoluted clauses. These conspire to degrade the clarity of the narrative.

The abstract needs to be completely rewritten, and there is ample room for improvement in all other sections as well. This effort will enable readers to better appreciate the exciting science.

We express our gratitude to the referee for their valuable comments. In response, we have revised the abstract, enhancing the clarity and comprehensibility of the manuscript, thereby facilitating the interpretation of the data.

What is the Fe and flavin stoichiometry of each of the preparations? This is crucial to interpretations of the mutations and deletions.

The iron and flavin content of the wildtype enzyme and all the variants is now given in the manuscript. All values are as expected and support our previously drawn conclusions.

What is the experimental evidence demonstrating which of FAD/FMN is the site of bifurcation?

The mechanism of electron bifurcation in StnABC, and also HydABC, is fundamentally different from classical bifurcation reaction as seen in NfnAB. The difference lies in the affinity modulation of the binding nucleotides and in kinetic gating rather than using the concept of crossed potential. There is no bifurcating FAD/FMN in StnB. Instead, the mechanism bears resemblance to the earliest discovered bifurcation complex, the cytochrome bc₁ complex (complex III) in bacterial and mitochondrial respiratory chains, which achieves electron bifurcation through the mobile Rieske domain. This is however not a main focus of this study and has been described in greatest detail in our recent work on the electron bifurcating hydrogenase.

Moreover, our findings provide compelling evidence that the FAD site in StnC does not serve as a bifurcating flavin. Instead, we demonstrate that the StnAB subunit, which shares homology with the HydBC subunit of the HydABC complex, is the actual site of bifurcation.

- (1) Our structural analysis reveals that the Arg residue in StnC's FAD is unable to effectively favour the formation of an highly reducing, short-living anionic semiquinone due to the relatively large distance (~6 Å) between the N5 nitrogen in the isoalloxazine ring of FAD and Arg. This distance

contradicts what has been observed in the case of NfnAB, where a closer proximity is necessary for successful bifurcation.

- (2) The StnAB subunit exhibits homology with the HydBC subunit of the HydABC complex. In our previous study by Katsyv et al., we presented a novel bifurcation reaction involving FMN in HydB, wherein the arrangement of nearby clusters triggered a state switch in FMN. Notably, we demonstrated that the deletion of cluster B2 in HydABC resulted in a complete loss of bifurcation activity. In our current investigation, we conducted experimental validation by introducing a mutation in the A1 cluster of StnA, which is homologous to HydC. This mutation led to the total abrogation of bifurcation activity, confirming the proposed role of the StnAB site as the actual site of bifurcation.

Why do the authors think that FMN switches between exergonic and endergonic electron transfer modes? The two transfers involved in bifurcation must be tightly coupled, which is to say neither occurs alone. Thus there is a single reaction involving both transfers.

The mechanism of electron bifurcation in StnABC is not a classical FBEB mechanism (see above). It uses redox-driven conformational changes as an alternative to kinetically gate the electron transfer. Therefore mobility of StnA and the switch between exergonic and endergonic electron transfer mode is essential for the mechanism that has been described and validated in detail in our previous publication.

Although the novel bifurcation mechanism is not the primary focus of our current study, we appreciate the referee's interest in obtaining a more detailed and extended description. However, it is important to note that the mechanism of StnAB is not at the core of the manuscript. The main emphasis of our manuscript lies in highlighting the remarkable modularity exhibited by the StnABC complex, which incorporates the StnAB bifurcation unit. For the referee we outline the key findings of the redox-driven bifurcation process below:

In our previous publication (Katsyv et al., 2023), we proposed a unique model for electron bifurcation in the HydABC complex, which is likely conserved in the StnABC complex as well.

The proposed mechanism involves the following steps:

1. First Round of NADPH Oxidation: During the initial round, StnC transfers two electrons in a stepwise manner to the FMN in StnB. The FMN, in turn, transfers these electrons to the B2 and A1 clusters while cycling through a one-electron reduced flavo-semiquinone state. Importantly, the reduction of the B2/A1 clusters significantly enhances the affinity of NAD⁺ binding on top of the FMN. This phenomenon has been demonstrated by Katsyv et al. using MST experiments and MD simulations. The bound NAD⁺ cation subsequently increases the redox potential of the flavin, enabling a two-electron reduction to form FMNH⁻ during the second round of NADPH oxidation. The FMNH⁻ state then transfers the hydride to NAD⁺ to generate NADH. The oxidation of two molecules of NADPH leads to the reduction of the B2/A1 cluster in the first round and the generation of NADH in the second round.
2. Blocking Electron Transfer to Fd: In the FMN-NAD⁺ bound state of StnABC, a highly stable reduced state of NADH/B2(red)/C1(red) is observed. This state effectively prevents electron transfer for reducing Fd, which binds at the C-terminus of StnB due to the relatively long distance between the B2/A1 and B3 clusters. We propose that the dissociation of the reduced NADH generates sufficient free energy to induce an outward conformational change in StnA (or HydC), closing the gap between the A1 and B3 clusters and facilitating efficient electron transfer. Similarly, the electron from B2 is shuttled to the B3/B4 cluster via the A1 cluster, ultimately leading to the production of two reduced Fd molecules. While the cryo-EM structure with an open HydC conformation has been resolved by Katsyv et al. and Feng et al., a similar conformation could not be resolved in StnABC due to constraints imposed by cross-linking. However, our mutational analysis targeting the A1 cluster, resulting in the complete loss of bifurcation activity, provides support for the mechanism proposed by Katsyv et al.

It is important to note that although the detailed mechanism described here extends beyond the main focus of our current study, we appreciate the opportunity to provide this extended description to address the referee's interest in the topic.

There is considerable confusion regarding whether bifurcation requires a stable semiquinone or a high-energy semiquinone. These are mutually exclusive so please determine which applies and explain how it can participate in bifurcation. For an exceptionally clear treatise, please see Nitschke and Russel, reference 10.

The Nitschke and Russel review describes a common mechanism for electron bifurcation where the flavin employs the properties of crossed potentials. Here the flavin splits the electrons in such a way that the first electron leaving has a more positive redox potential than the second one. Interestingly, the flavin takes advantage of a highly negative one-electron couple to drive the reduction of the low-potential acceptor, ferredoxin. However, this entire condition is absent in the case of FMN in StnB (or HydB), where no flavin-based electron bifurcation (FBEB) takes place (see above).

Currently, there are three available structures of the HydABC complex; however, none of them have successfully determined the reduced states of the FMN, which plays a pivotal role in a novel bifurcation process akin to the StnABC complex. It is worth noting that spectroscopic investigations of the related HydABC complex by the renowned Birrell group have provided indications of a neutral semiquinone (NSQ) state in the FMN of HydB. However, it is important to acknowledge that these spectroscopic measurements are highly intricate and require specialized expertise, which falls outside the scope of our current study's revision.

It is worth mentioning here that Katsyv and co-workers used DFT calculations and MD simulations to propose that FMN forms one electron-reduced flavo-semiquinone state in the first round of reduction. Whereas after NAD⁺ binding, the redox potential of the FMN increases, which could favour a hydride transfer.

Please make a clear consistent distinction between bound cofactors (prosthetic groups) and substrates/products. The cofactors (prosthetic groups) are those that remain bound to the enzyme: FeS clusters, Zn, FMN and FAD, whereas NAD(H) and NADP(H) have the status of substrates and products in this reaction. Even Fd is technically a substrate/product in this case.

We apologise for the confusion, we have now added a table (Table 3) in the supplementary to a clear distinction between prosthetic group and substrates.

Please list the complement of cofactors (=prosthetic groups) present in the 'oxidized' samples, for comparison with those present in the reduced samples. In its current form, the manuscript says that cofactors are absent from the oxidized protein. If it is genuinely apoprotein then it is no wonder that there is increased disorder in some places and displacement. Hopefully only NAD(H) and NADP(H) are absent? Moreover if simply not adding them to the buffer results in their absence from the structure, this will confirm that they are relatively weakly bound and thus that they do not have the status of cofactors. Maybe there is an issue in translating terminology to English, but we understand that a cofactor assists in catalysis. NAD(H) and NADP(H) are certainly recycled in cells, but for any given reaction, each is either a substrate or a product. (Exceptions include cases where a pyridine nucleotide is an allosteric effector.)

We have now added a table to distinguish between the cofactors present in the oxidised and the reduced state (Table 2 and 3).

We would like to highlight that in the oxidized (apo) state, we only observed density for the FAD, while FMN was absent. This absence of FMN in the apo state is likely attributed to its loose binding. A more

comprehensive explanation regarding the lack of FMN in the apo-structure is provided in our response to referee 2.

A claim is made that because a portion of the protein is disordered, the clusters it contains cannot be important for catalytic electron transfer. Why? Could the disorder represent static disorder stemming from partial occupancy?

The C-ter domain containing C1 and C2 [4Fe4S]-clusters is not disordered in the StnC subunit from *S. Ovata*. The C1 and C2 are not part of the electron transfer pathway as this branch is a “non-functional” evolutionary remnant in terms of electron transfer. The region is homologous to the FdhF from *Desulfovibrio gigas* with an RMSD values of 3.2 Å, and forms large interactions in the tetrameric interface of the Stn assembly possibly to provide extra stability. Even in HydABCSL structure the A4 and A5 [4Fe4S]-cluster do not participate in the main chain electron transfer from Ni-Fe centre in HydL subunit to FMN-NAD⁺ binding site in HydB.

More recently, the homologous NfnABC complex, akin to the StnABC complex, has been successfully purified from *P. furiosus*. We conducted a comparative analysis of the overall subunit architecture of StnC with the alphafold models of the NsoC and NfnA subunits. While significant portions of the subunits displayed a similar architecture to the StnC subunit, the C-terminal FdhF domains of both NsoC and NfnA subunits were found to be disordered and lacked the binding motif for the C1 and C2 clusters. This observation further supports the notion that the C1 and C2 clusters do not participate in electron transfer in NsoC and NfnA.

Please define all abbreviations at the time of first use.

Done.

How was it determined that the Zn is 'strongly' bound? bond lengths will not be precise enough to arbitrate at this resolution.

The referee has appropriately pointed out that the resolution of our cryo-EM structures does not allow for precise measurement of bond lengths involving the bound Zn²⁺ ion and its ligands in the StnB subunit. Nevertheless, we want to emphasize that we observed strong electron density for the Zn²⁺ ion in our cryoEM structures (Fig. S4) and have quantified stoichiometric amounts of Zn²⁺ using ICP-MS analysis as part of the complex characterization. Furthermore, it has been convincingly demonstrated by Katsyv et al. and Furlan et al. that Zn²⁺ binds to the C-terminus of the HydB subunit (which is homologous to StnB) and plays a role in stabilizing this region.

What are the oxidation states of the bound cofactors (prosthetic groups) of the 'OX sample' and how do they compare with the same in the reduced sample? Please at minimum provide optical spectra of each and a difference spectrum (scale the two parent spectra to the same enzyme concentration before subtracting). The 'Methods' says that protein complex without any FMN, NAD⁺, NADPH or Fd was used for the 'Oxidized' sample. In what respect is it oxidized? Was it exposed to air?

We apologize for any confusion caused. In our study, the term "apo state" refers to the oxidized structure of the StnABC complex, where no exogenous FMN, NAD, NADPH, or Fd was introduced prior to cryo-EM grid freezing under anaerobic conditions (95% N₂ and 5% H₂). It is important to note that the sample was not exposed to air during the experimental process.

In contrast, the reduced state of the structure corresponds to the FMN-NAD⁺ bound state. In this state, NADPH, as a natural reducing agent, was used to release electrons within the complex, resulting in the reduction of the bound FeS cluster. FMN and FAD was added prior cryoEM experiment to compensate for any loss of flavin, while NAD⁺ was introduced to investigate its binding to the FMN within the StnB subunit.

The determination of specific oxidation and reduction states of the bound cofactors is difficult and out of the scope of this study. Given that the tetrameric StnABC complex comprises numerous FeS clusters (48 in total), as well as multiple FAD/FMN and NAD⁺ molecules, it is not feasible to assign individual states to each cofactor due to potential signal overlap. We have sought guidance from experts in the field, all of whom confirm the inherent difficulty in spectroscopically investigating the states of both the StnABC and HydABC complexes. Their consensus is that such determination is indeed unattainable.

Given that protein was crosslinked prior to deposition on grids, we do not expect to see extensive conformational sampling. Correct? The text nevertheless refers to 'dynamics'. Would it be better described as disorder? That would lower the resolution of the region in question but represent static disorder rather than dynamics. It could mean that the local structure is not strongly constrained to a unique conformation, rather than that individual regions of protein move back and forth on an ongoing basis.

Prior to vitrification, cross-linking is a widely adopted standard procedure in the field, commonly employed for highly dynamic transcription and chromatin remodelling complexes. While the captured states may not perfectly replicate those observed in non-cross-linked samples, they provide valuable insights into the proximal conformations. In this context, we can draw comparisons to our previous study on the closely related HydABC complexes.

Furthermore, although global dynamics are constrained to a single conformation, it is important to note that local dynamics can still be inferred. For instance, the binding of NAD⁺ on top of FMN is contingent upon the reduction of the B1 and A1 [2Fe2S]-clusters, as demonstrated by our FMN-NAD⁺ bound (holo) structure. Additionally, the observed dynamics in the loop region (residues 182-191) of the StnB subunit are attributed to the binding of NAD⁺ rather than FMN, as elucidated by Katsyv et. al. Therefore, it is more appropriate to refer to these changes as local dynamics rather than disorder, as the latter may evoke a different interpretation in the readers' minds.

It is unclear what evidence forms the basis for the proposed sequence of events in the mechanism. Moreover the mechanism needs a much clearer description please.

We apologize for not including a detailed explanation of the mechanism. We have now included a detailed description of the mechanism and have explained how our mutational analysis support the mechanism reported by Katsyv et. al.

Considering that the structural resolution is 3 Å, and that is itself substantially based on modeling, it is premature to discuss subtle orientational details, such as the orientation of the terminal guanidinium of an Arg.

The overall resolution of the structure is 3 Å, while the core region exhibits a higher resolution of 2.5 Å. This improved resolution enables us to model the core region of the complex with greater precision compared to other regions with lower resolution. Since the specific Arg residue mentioned by the referee is situated within the core region, we can model its orientation and approximate distance from FAD with reasonable confidence. Furthermore, the electron density maps and models undergo scrutiny by other referees, who validate the accuracy and reliability of our findings.

Please specify what operations were conducted under inert atmosphere. Readers need to be told this, as many will not be familiar with air-sensitive enzymes.

We have included a detailed descriptions in the methods section.

For the Bradford assay, what protein was used for the standard curve?

We used BSA for the standard curve. We included the information in the methods section.

Caption of Figure 5: the text implies that both electrons from NADPH travel down a chain of Fe₄S₄ clusters, but FeS clusters are normally carriers of single electrons. Please clarify what is being proposed.

We have now included a detailed explanation of the mechanism, which clarifies the doubt.

Figure captions in general lack needed detail.

Legend in Figure S9, there is a typo in 'So StnB Redcued' (should read 'Reduced').

We have corrected this typo.

Reviewer #2 (Remarks to the Author):

What are the noteworthy results?

CryoEM structures of StnABC from *Sporomusa ovata* along with mutagenesis and activity assays to probe mechanistic features. See attached review for details.

Will the work be of significance to the field and related fields?

Yes. See attached review for details.

How does it compare to the established literature? If the work is not original, please provide relevant references.

The work is highly original. See attached review for details.

Does the work support the conclusions and claims, or is additional evidence needed?

Overall, yes, but I recommend some additional discussion to clarify some points. See attached review for details.

Are there any flaws in the data analysis, interpretation and conclusions? Do these prohibit publication or require revision?

There may be some minor flaws. See attached review for details.

Is the methodology sound? Does the work meet the expected standards in your field?

For the most part, but there are some minor questions on this that I would like to see addressed. See attached review for details.

Is there enough detail provided in the methods for the work to be reproduced?

Yes.

The manuscript "Molecular architecture and electron transfer pathways of the Stn family transhydrogenase" by Kumar, Kremp and coworkers reports cryoEM structures of the electronbifurcating transhydrogenase from *Sporomusa ovata* (StnABC) in an "oxidised" and a "reduced" state. The authors find that the enzyme forms a dodecameric structure comprising four functionally independent StnABC heterotrimers. The structure and arrangement of cofactors is similar to other related electron-bifurcating enzymes, and a similar mechanism of electron-bifurcation at the FMN cofactor in the StnB subunit is proposed. Interestingly, changes in the FAD and proximal [4Fe-4S] cluster environment in the StnC subunit are proposed to be responsible for changing this flavin from an

electron-bifurcating form in the related Nfn complex to a non-electron bifurcating Flavin in StnABC. The authors were also able to produce the enzyme heterologously in *E. coli* and used this system for targeted mutagenesis. Overall, I find this to be an interesting and well executed study and I can recommend publication after consideration of some minor points and slight clarification of the text.

We thank the referee for the positive assessment of our work and providing insightful/valuable suggestions on our work.

The lack of FMN in the “oxidised” enzymes has not been adequately addressed. At the top of Page 5 it is stated that:

“For reducing the enzyme, a mixture of cofactors was added to initiate the reaction, followed by immediately vitrifying the grids (see methods). The structure under oxidizing conditions was obtained by adding no cofactors.”

In the methods, this is specified to mean that FMN, NAD⁺, Fd and NADPH were added or not. Table 2 implies that the oxidised enzyme lacks FMN entirely and at the top of Page 6 it states that: “The structure solved under oxidizing conditions only contained FAD in the StnC subunit, while under reducing conditions FMN-NAD⁺ was also found to be present in the StnB subunit.”

If the oxidised enzyme lacks FMN then how is it able to perform catalysis? The activity assays reported on Page 4 do not appear to have been carried out in the presence of exogenous FMN.

This is a valid point raised by the referee. We would like to point out that probably a small fraction of the StnABC complex is purified with bound FMN, which helped the complex catalysing the bifurcation reaction. Adding exogenous FMN would increase the bifurcation activity as more protein complexes would harbour the FMN, which is reported by Kremp et. al. 2020. This behaviour has also been shown by Schuchmann et. al. (ref 13) and Feng et. al. (ref 18) with Fe-Fe and Ni-Fe hydrogenase HydABC complex, respectively, where adding exogenous FMN yielded higher bifurcation activity. Moreover, in our previous publication (Katsyv et. al. ref 17), we did not include exogenous FMN in our purification, activity, or MST measurements, and complex could still catalyse bifurcation activity.

More generally, it is not clear why the oxidised enzyme lacks FMN, why the authors did not deem it sensible to attempt reconstitution of FMN for their oxidised structure, and whether the differences between oxidised and reduced enzymes simply reflect the lack of FMN. In the methods it is explained that 5 μM FAD was included in the protein purification buffers, presumably to counteract loss of FAD. Why was FMN not also included? On Page 9 it is stated that:

“The comparison of StnABC structures in the oxidized and reduced state revealed no major differences, however, we could see the loop (residues 182–191) of StnB undergoing a significant conformational change upon NAD⁺ binding (Fig. 4e).”

How do the authors know that this change is not simply due to binding of FMN? Overall, I think the authors should be more transparent about the lack of FMN in the oxidised structure and rationalise why they observe enzyme activity for this preparation e.g. with reference to the higher catalytic rates observed in the presence of exogenous FMN reported in reference 12.

Due to the observation that the complex retained its bifurcation activity even without exogenous FMN, we proceeded with determining the structure without the addition of exogenous FMN. In the apo structure, we only detected the presence of FAD and not FMN. This suggests that FMN is only a loosely associated cofactor and is not bound to the majority of the protein complex, where it is averaged out in the single-particle cryo-EM structure.

Our initial experiments were based on Kremp et al., in which the StnABC complex was proven to contain FAD, whereas FMN was not detected (Fig. S11 in ref. 12). This is the reason why we focused on purifying and including only FAD in our experiments to maintain consistency. For this apo or oxidized state of StnABC, we did not include exogenous FMN, NAD⁺, and NADPH. The structure of the apo StnABC complex,

particularly when comparing the StnB and HydB subunits, is similar to the apo structure of the HydABCSL complex reported by Feng et al.

This apo structure of StnABC provided us with guiding information indicating that the StnAB module is responsible for bifurcation, which is consistent with the findings of Katsyv and Feng et al. To further investigate the structural binding of FMN and NAD⁺ in StnB, we conducted additional cryo-EM experiments adding 10 μM FMN, 500 μM NAD⁺, 30 μM Fd, and 500 μM NADPH. In this setup, NADPH served as a natural reducing agent to reduce the complex. The electrons would be transferred to FMN and subsequently translocated to the B1 and A1 clusters. The reduction of B1 and A1 triggers the binding of NAD⁺ to FMN, as demonstrated by Katsyv et al. Our holoenzyme with FMN-NAD⁺ bound in StnB subunit proves that the StnAB module performs bifurcation as reported for HydBC. Reconstituting FMN in our apo structure would have still left us with two unanswered questions: (1) which flavin is involved in bifurcation, and (2) whether the StnAB module performs bifurcation similarly to HydBC. To address these questions and avoid redundancy, we conducted direct experiments with all exogenous FMN, NAD⁺, Fd, and NADPH.

In Katsyv et. al. we managed to determine HydABC structure with only FMN bound. Comparing the apo vs NAD⁺ bound structures of HydABC revealed the loop in HydB (residue 197-206) sampled in different conformations. Moreover, this claim was supported by extensive MD simulations on apo vs NAD⁺ bound state. Therefore, the loop dynamics in HydB is indeed associated with the binding of NAD⁺ at FMN, and not binding of FMN alone. We compared the apo vs FMN-NAD⁺ bound state in StnB, and observe similar dynamics associated with the loop (residue 182-191).

On the topic of catalysis, I was struck by the observation that NADPH-dependent ferredoxin reduction (1.59 U/mg) and NAD⁺ reduction (2.80 U/mg) do not have the same rates. The authors should clarify the reason for this. It cannot be due to reduction of only one of the two iron-sulphur clusters in ferredoxin (indicated in Figure 5), as the rates are calculated based on "One Unit is defined as the transfer of 2 μmol electrons/min." Thus, it would appear that 36% of electrons go to ferredoxin and 64% of electrons go to NAD⁺. Is there some ferredoxin:NAD⁺ oxidoreductase activity? Could the authors clarify this in the current manuscript with reference to the activity measurements reported in reference 12? From a brief look at reference 12, it appears that there is both a ferredoxin-independent NADPH:NAD⁺ oxidoreductase rate and a Fdred:NAD⁺ oxidoreductase rate, which presumably distort the stoichiometry. Related to this, could the authors explain how they monitor NADH and NADPH independently even though they have the same absorbance spectrum. Presumably nucleotide-specific regeneration systems are used and assumed to keep the absorbance or either NADPH or NADH constant as reported in reference 12?

The Reviewer is right. In general, the rates of Fd and NAD⁺ reduction should be the same since NADPH is oxidised and the electrons are split up to reduce Fd and NAD⁺ at the same time. In Reference 12 it is already reported that the Stn not only has a Fdred:NADP⁺ oxidoreductase activity which pushes the ratio towards NAD⁺ reduction, but also a NADPH:NAD⁺ oxidoreductase activity that does the same. Therefore, to us, it was not surprising to see that the ratio deviates from the expected 1:1 (Ferredoxin reduced:NAD reduced per NADPH oxidised).

The Reviewer is Right again, wherever possible we used regeneration systems to keep the nucleotide (NADPH) pool in the reduced state:

The enzyme assays were performed as it is stated below and in reference 12:

Simultaneous reduction of ferredoxin and NAD⁺ with NADPH. Reduction of ferredoxin and NAD⁺ with NADPH catalysed by the Stn complex was assayed at 50 °C in 50 mM Gly–Gly buffer (pH 8) containing 10 mM NaCl and 20 mM MgSO₄. To keep the level of NADPH constant, 0.25 mM NADP⁺ were pre-reduced with 1 unit glucose-6-phosphate dehydrogenase and 20 mM glucose-6-phosphate (NADP⁺ reducing system). 30 μM ferredoxin and 0.5 mM NAD⁺ were used as electron acceptors. To determine the reduction of ferredoxin and the reduction of NAD⁺ the absorbance was followed at 430 nm ($\epsilon_{\Delta\text{ox-red}} = 13.1 \text{ mM}^{-1} \times \text{cm}^{-1}$) and at 340 nm ($\epsilon = 6.3 \text{ mM}^{-1} \times \text{cm}^{-1}$), respectively.

In the case of the NADP reduction assay, we did not use an NADH Regeneration system and corrected the rate of NADP reduction by the rate of NADH oxidation (assuming a Ratio of 0.5 NADH is oxidized/NADP reduced):

NADP⁺ reduction with NADH and reduced ferredoxin. NADP⁺ reduction catalysed by the Stn complex was assayed at 50 °C in 50 mM Gly-Gly buffer (pH 8) containing 10 mM NaCl and 20 mM MgSO₄. 30 μM ferredoxin was pre-reduced with CODH using a CO-atmosphere. 0.25 mM NADH and 0.5 mM NADP⁺ were added and the change of absorbance at 340 nm was followed to determine NADP⁺ reduction. Since NADH oxidation leads to a change in absorbance at 340 nm as well, the rate of NADP⁺ reduction has been corrected by the NADH oxidation rate.

I was also struck by the lack of the plant-type ferredoxin [2Fe-2S] cluster in the N-terminal NuoG-like domain of StnC (equivalent to N1b in NuoG/Nqo3 and A4 in HydABC). Could the authors add a brief comment about this? The cluster binding motif appears to be present so it is surprising that there no evidence for this cluster in the structure.

We appreciate the referee for bringing this to our attention. We apologize for the oversight in not refining the [2Fe2S]-cluster (C4) in the NuoG domain of StnC. We have now addressed this issue by adding and refining the [2Fe2S]-cluster in the StnC subunit. Consequently, we have made appropriate revisions to the text and figures to reflect this update. As correctly pointed out by the referee, similar to the A4 cluster in the HydABC complex, the C4 cluster in StnC also plays a role in the main chain electron transfer pathway from FAD in StnC to FMN in StnB.

Minor points:

1. Abstract: “sustain” > “sustain life” – Done
2. Page 2: “adenosinetriphosphate” > “adenosine triphosphate” – Done
3. Page 3: “...ratio of NAD⁺/NADH is 30/1...” – ratios normally given with “:” – Done
4. Page 3: “Contrary...” > “On the contrary...” or “In contrast...” – Done
5. Page 3: “...anaerobes do neither have...” > “...anaerobes neither have...” – Done
6. Page 3: “...for long time.” > “...for a long time.” – Done
7. Page 3: “...NADPH to NAD drives...” > “...NADPH to NAD⁺ drives...” – Done
8. Page 3: “a semiquinone anion is stabilized resulting in two very different redox potentials (“crossed potentials”) that allow one-electron transfers to acceptors with high and low potentials 10.” This should be clarified. It is possibly simply a typo but I think “stabilized” should be replaced with “destabilized”. In my understanding of reference 10 “crossed potentials” indicates that the redox potential of the semi-reduced/fully-reduced pair is more positive than the redox potential of the oxidised/semi-reduced pair, meaning that the semi-reduced state is destabilized. Additionally, on Page 9 it is stated that “The FMN coordination is different from other bifurcating complexes which have an arginine or lysine to stabilize an ASQ for using the concept of cross potential where a flavin stabilises the ASQ using an Arg/Lys residue to release two electron on spatially separated pathways with very different potentials (Fig. S9).” Again, I don’t think that stabilization of the ASQ leads to crossed potentials.
We have now clarified our statements in the manuscript.
9. Page 3: “In addition to the bifurcating flavin, a second flavin is required to collect the single electrons before a hydride can be transferred to NAD⁺ or NADP⁺.” While this is indeed the case for Nfn and a number of other FBEB enzymes, it is not the case for Sfn or HydABC. Perhaps the word “usually” should be inserted between “is” and “required”? – Done
10. Page 3: “...a comparable simple...” > “...a comparably simple...” – Done
11. Page 3: “...NfnAB.” > “...NfnA and B.” – Done
12. Page 4: “...low potential FADH^o- radical...” > “...low potential FADH[•]- radical...” – Done
13. Page 4: “Interesting, Stn has...” > “Interestingly, Stn has...” – Done
14. Page 5: “All cryoEM grids were prepared in a strictly anoxic chamber with a gas composition of 95% N₂ and 5% H₂, i.e., under catalytic turnover conditions.” Why do these conditions imply catalytic turnover? Aren’t nucleotides and ferredoxin are required for this?

Thanks for pointing out, catalytic turnover condition is indeed obtained when nucleotides are introduced. We have now rephrased the sentence.

15. Page 5: "...using AlphaFold..." – here it would be appropriate to cite reference 44. – Done

16. Page 5: "The tetramer comprises of four monomeric StnABC heterotrimeric complexes..." – 1. Delete "of", 2. What is a monomeric heterotrimeric complex? Surely by definition a heterotrimer is trimeric? Perhaps rephrase this for clarity.

We would like to keep it that way. Heterotrimeric – trimer of different proteins, whereas trimer – same protein.

17. Page 5: "...six [4Fe-4S] cluster..." > "...six [4Fe-4S] clusters..." – Done

18. Page 6: "...degenerate [Fe]-only hydrogenase..." What is meant by degenerate? Also, the term [FeFe]-hydrogenase is used on Page 4. If the authors wish to refer to the same class of enzymes then I recommend using the same nomenclature.

Sorry for the mistake, we have removed the word degenerate from the sentence.

19. Page 7: "...simultaneous reduction of NAD⁺ and Fdox with NADP⁺..." Presumably NADPH was meant? – Done

20. Page 7: "...the metals the [4Fe-4S] cluster..." > "...the metals of the [4Fe-4S] cluster..." – Done

21. Page 8: "...electron-bifurcating hydrogenases from *Thermoanaerobacter kivui*..." Are there more than one?

Sorry, there is only one HydABC in *T. Kivui*. We have changed the word to hydrogenase.

22. Page 8: "...which represents the binding site for Fd reduction..." Was this proven? I thought it was just a hypothesis.

Katsyv et. al. has shown the putative binding of Fd reduction at the C-ter domain of the HydB subunit using MD simulations. They also truncated the C-ter domain of HydB, which resulted in severe reduction of bifurcation activity. Therefore, the Fd reduction site should be at C-ter of HydB. However, we have added the word putative in the sentence - "...which represents the putative binding site for Fd reduction..."

23. On Page 9 it is stated that: "Overall, we expect FMN to go through a transient one-electron reduced flavo-semiquinone state, which has also been reported for HydABC 21." Reference 21 does not report any evidence for a flavo-semiquinone state. This reference proposes the existence of one. In a supplementary figure (S11) from Chondgar et al., (2020) JBIC some evidence for an FMN radical can be observed in EPR spectra of HydABC in the presence of NAD⁺.

Thanks for pointing it out, we have cited the paper.

24. Page 11: "The described mechanism differs drastically from conventional FBEB enzymes 11,19,24-27 and is rather analogous to the bifurcation reaction that was initially described as a part of the Q-cycle in complex III (cytochrome bc₁ reductase) within respiratory chains 28,29." In what way is the electron-bifurcation reaction of StnABC analogous with the Q-cycle in complex III?

The StnAB module performs a novel bifurcation reaction similar to the one reported for HydBC. The method of redox driven conformational change as an alternative to the concept of crossed redox potential is known for the cytochrome bc₁ reductase. In complex III (cytochrome bc₁) in bacterial and mitochondrial respiratory chains achieves electron bifurcation via the mobile Rieske domain. Similar to the StnA [2Fe-2S] cluster motion, the [2Fe-2S] cluster of the Rieske domain in complex III employs a kinetically controlled bifurcation mechanism to achieve electron shuttling between the low-potential b-heme and the high-potential c-heme in the Q-cycle. The Rieske domain movement is dependent on the redox state of the [2Fe-2S] cluster.

25. Page 11: "...the GltD domain of the Stn complex has evolved an FAD-binding pocket that allows electron bifurcation..." Is Nfn meant here?

No, the GltD domain is from the glutamate synthase of *E. coli*. They are known for binding to NADPH, but do not contain a FAD binding pocket. On the other hand, NfnB and StnC also contain this particular domain with similar NADPH binding site, in addition to the FAD binding pocket.

26. Page 12: "...favours electrons transfer..." > "...favours electron transfer..." – Done

27. Page 12: "...may represent a hydrogen species, e.g., a proton..." Is this realistically observed in cryoEM density at this resolution? Is the density attributed to other nearby protons distinguishable? I would rather favour the idea of a water molecule or change to the cluster geometry.

It's only a hypothesis which we speculate based on electron density observed for residue K170 and C114 under oxidising and reducing conditions. We cannot see any water molecule in our cryo-EM density.

28. Page 12: "...in the enzymes oxidised and reduced condition..." > "...in the enzyme's oxidised and reduced condition..." – Done

29. Page 12: "In summary, our study highlights how nature has come up with two solutions for electron bifurcation in the GltD domain." But I had understood that the GltD domain in StnC was not electron bifurcating? Please rephrase this sentence.

Thank you. Paragraph was rephrased for clarity.

30. Figure 2 caption: "...one [2Fe2S] clusters..." > "...one [2Fe2S] cluster..." – Done

31. Figure 2 caption: Ångström appears to have an Å (with an accent) rather than an Å – Done

32. Figure 3 caption: "...two Rossmann fold..." > "...two Rossmann folds..." – Done

33. Figure 4f: a slightly unconventional nomenclature is used for the iron-sulphur clusters in Nqo1-

3. I recommend using the nomenclature reported in reference 22, Figure 1B. Also, it isn't clear why this particular orientation of the cofactors in Nqo1-3 has been chosen as it doesn't align well with the related views from StnAB and HydBC. Furthermore, clusters have been included in the Nqo1-3 view that have related clusters in StnAB and HydBC

We have now modified the figure.

34. Figure 4 caption: "...nearby A1 and B2 cluster..." > "...nearby A1 and B2 clusters..." – Done

35. Throughout: please harmonise the iron-sulphur cluster nomenclature. Sometimes [4Fe-4S] is used, sometimes [4Fe4S] and sometimes 4Fe4S. Same for [2Fe-2S]

Thanks for pointing this out, we have now harmonised the nomenclature.

36. Throughout: a mixture of english english (e.g. coloured, homologue) and american english (e.g. colored, homolog) is used. Please harmonise. – Done

Reviewer #3 (Remarks to the Author):

Anuj Kumar et. al present the cryo-EM structure of Stn class transhydrogenase StnABC complex from *Sporomusa ovata* showing a tetramer of trimeric arrangement. The authors demonstrate its unique structure and function characteristics by necessary functional experiments and structure comparison with the known NfnAB and HydABC hydrogenase structures.

The text and figures are clear. The methods and results, including cryo-EM, are sound. I recommend this manuscript for publication in Nature Communications

We thank the referee for appreciating our work and recommending it for publication in Nature Communications.

I have a suggestion for the authors:

The relevant experiments were conducted under strictly controlled conditions (reduction or oxidation), and structural and functional analyses suggest that a single protomer is the minimal functional unit. These imply that the structure of different protomers should be uniform. However, it is always worth using symmetry expansion and/or local refinement (all available in cryoSPARC and Relion) to investigate the "personality" of protomers in an oligomerization complex to see whether the complex is indeed symmetrical or pseudo-symmetric.

This is an excellent suggestion by the referee. We did consider the option of performing symmetry expansion and refinement using C1 symmetry to investigate the structural dynamics and conformational changes between the oxidized and reduced states of the enzyme. However, due to the presence of cross-linking in both conditions, all four protomers of the enzyme are constrained to a single conformation. Consequently, it is not possible to visualize the structural dynamics associated with individual protomers. However, we were able to infer local dynamics, such as the binding of FMN and NAD⁺, in the reduced state of the enzyme (with NADPH), as these processes do not require significant conformational changes of the

subunits. Additionally, the occupancy of FMN and NAD⁺ was observed in all four protomers even when refined with C1 symmetry or through symmetry expansion followed by local refinement. In summary, our cryo-EM structures exhibit symmetry when refined using C1 or D2 symmetry, but each protomer functions independently, as the distances between the nearest cofactors in adjacent protomers exceed the physiological range for electron transfer.

Reviewer #4 (Remarks to the Author):

Short transhydrogenase (Stn) is a recently discovered flavin-based electron bifurcation (FBEB) enzyme that belongs to the HydABC superfamily. Kumar et al. reports the enzyme structure and a structural-based mutagenesis and activity assay. The work reveals the enzyme architecture, electron transport paths, and several unique features, leading the authors to propose a plausible electron bifurcation mechanism. The manuscript is well written, but we have a few concerns as detailed below. Overall, this work is a valuable contribution to the field.

We thank the referee for the positive assessment of our work. We have clarified the concerns raised in below points.

Major concern:

1. The states of the samples need to be better explained. Both oxidized- and reduced-state samples were crosslinked and prepared in an anaerobic chamber. The difference was that the authors added additional cofactors to the reduced state. Why was the other sample without additional cofactors assigned oxidized state? Can you compare the structure of this sample with an air-oxidized sample?

We thank the referee for pointing this out and apologize for the confusion. We have now better clarified the oxidised and reduced states of the enzyme. The oxidised state of the enzyme is the apo form where NO exogenous substrates and products were added. We termed the apo state as oxidised as no additional reducing agents (NADPH) were introduced. Whereas the reduced state is the holoenzyme with FMN-NAD⁺ bound (holo state) obtained by introducing exogenous FMN, FAD, NAD⁺, Fd, and NADPH.

Theoretically, the apo structure of the StnABC complex should be similar to the sample exposed to oxygen. However, due to the risk of denaturation of highly oxygen-sensitive metal clusters, it is recommended to avoid air exposure of the sample. Moreover, all purification steps and activity assays were performed anaerobically. Under an oxygenic environment, the protein is inactive.

2. The authors have solved the StnABC structure as a tetramer. But in Fig. S7 panel d, the WT StnABC was a monomer and dimer in the solution according to mass photometry measurements. Can you explain the observed monomer and dimer states? Is it possible that the mild crosslinking fixed the complex into an artificial state? If so, they should determine the structures of the uncross-linked monomer and dimer which are large enough for cryo-EM.

The StnABC intrinsically remains an intact tetrameric complex at high concentrations (0.2 μ M or higher) even before crosslinking, as seen in the size exclusion profile (Fig. S1b and Kremp et. al. 2020). The crosslinker would provide extra stability to the tetrameric complexes in solution to withstand the shear force and air-water interface during cryo-EM grid preparation. We had initially plunged the cryo-EM grid with non-crosslinked samples, however, most of the complex was destroyed and heavily aggregated due to the air-water interface. It was not even possible to obtain reference free 2D class averages.

The mass photometry measurements were conducted with non-crosslinked samples at a protein concentration of 25 nM, which is very low compared to the amount of protein used in size exclusion (2.4 μ M) or cryo-EM (1.2 μ M). Mass photometry is usually performed with low amounts of protein concentration to record single molecule events for precise determination of the mass. Due to the dilution effect at low concentrations, it is expected that the tetrameric StnABC complex will dissociate into monomers and dimers which we see in the mass photometry readouts.

3. The EM density map in the core region of the complex looks decent. But the local resolution of several other regions, such as StnA, the N-terminal domain, and the C-terminal domain of StnB are difficult to assess due to the large map size, as these regions may not have reached 4 Å resolution, the upper color display bound used in the local resolution figures. The authors should revise these figures to better show these regions and with a color range higher than 4Å. The authors should also explicitly acknowledge that these regions do not allow atomic modeling but can fit the AlphaFold models.

We have now revised the local resolution panel in Fig. S2 and S3, where the upper limit is higher than 4 Å. Moreover, we have explicitly stated in the text and Fig. S4 legend that precise atomic modelling of the N- and C-ter of StnB and C-ter of StnA was not possible due to high flexibility resulting in weak electron density.

4. The EM density of all cofactors should be presented (if not already shown in Fig. S4 or other figures). The authors should improve the modeling of NADP: the model does not fit the density well, and there are many severe clashes with the surrounding protein residues.

We thank the referee for pointing this out. As the NADPH only transiently binds to the StnC to release electrons at FAD site, a very weak density for NADPH was observed in the reduced structure. Therefore, we have now modelled the NADPH binding based on the crystal structure of the NfnAB with bound NADPH. All the clashes surrounding the area have been resolved.

Minor:

1. Results section Purification of the Stn complex, please highlight Methyl Viologen when first mentioned with MV since the authors used the initials in the rest of the manuscript.

This has been corrected.

2. Fig. S7 panel c the SDS-PAGE is not discussed in the legend.

We have added a sentence.

3. There are two D471 labels in panel C of Figure 3.

Done

4. Discussion page 11, "Our study demonstrates, for the first time, that the hydrogenase domain of HydABC can be functionally replaced with another redox enzyme...". This is not the first structural evidence. Structural available [NiFe]-hydrogenase ABCSL and HydABC have very different enzyme subunits coupled to the bifurcation module.

We apologize for the misunderstanding. While we acknowledge that [NiFe]-hydrogenase ABCSL and HydABC complexes are distinct entities, our point was to highlight the versatility of the BC subunit in forming associations with various enzymes other than a hydrogenase.

REVIEWERS' COMMENTS

Reviewer #1 (Remarks to the Author):

The results are certainly noteworthy and will be of significance to the field. The work is original and compares favorably with existing literature. Arguments, experimental support and methodology are sound, but chemical units are needed in the table that reports catalytic activities.

Methods and description of samples has been significantly improved in this revision. Several ambiguities and mis-statements have been corrected with the result that clarity is significantly improved.

Thank you for the effort this entailed.

This is much improved, so that even aspects that were good before, are able to shine. I note for example the glorious figures and the numerous assays of catalytic activity, performed in triplicate, which are such excellent complements to structural findings.

HOWEVER: no chemical units were to be found for any of the assays, and the Table 1 reports wavelengths at which observations were made in lieu of units. The wavelengths describe how the assay was performed, but they do not allow any quantitative conclusions to be made with the numbers in the table. Only comparisons between protein variants. Please provide SOMEWHERE the conversion between 'Units' and $\mu\text{moles/L}$ ($=\mu\text{M}$) limiting substrate consumed per $\mu\text{mole/L}$ concentration of enzyme per unit of time (presumably at saturating concentrations of all other co-substrates). If more than one (co)substrate is limiting then there will be additional units, but they should all be units of concentration ($\mu\text{moles/mL}$). Ideally, the enzyme concentrations should be in some molar quantity not mg/mL , so they provide chemical insight. Sadly the biochemical community fails often in this regard and the consequence is that readers do not realize just how good our favorite catalysts really are!

Several abbreviations are still not defined anywhere: Gltd, NuoE..... at least provide provide a glossary in the SI. Only insiders and nerds realize that modules of additional NAD(P)H-utilizing enzymes are implied, so this is not merely a good habit, it is required for readers to appreciate the significance of the author's finding of homology with domains of StnC. Lines 203-204 did a very good job in this regard.

Thank you for providing critical information on stoichiometry of bound cofactors, this greatly increases the value of the results. At the time of first reporting, please tell the reader the number of Fe expected

(42) instead of simply saying 'as expected', which omits the information needed for readers to assess your results (line 124).

Instead of the term 'oxidized', please call the untreated enzyme 'as-isolated'. No experimental determination has been made of oxidation state, so the vocabulary should avoid making unsubstantiated claims. Moreover this will remove an apparent claim to have oxidized the samples and maintained them as such during irradiation with a stream of electrons (!!). Since the samples were never actively oxidized (not even exposed to air), and there is not yet an agreed-upon way to keep these samples oxidized during EM, even if they started that way (which I very much doubt), the authors should not claim to have 'oxidized' material, anywhere.

Similarly, since there is no determination of the oxidation/reduction state, replace the terminology 'reduced' with '+NADPH'. This is a simple accurate statement that will be clear and honest. It completely obviates any responsibility to determine oxidation states and it allows the reader to know what is present in the samples.

The authors have convinced me that spectroscopy to make determinations of oxidation states is too much to ask, I agree, ON THE CONDITION that the authors cease making claims (throughout the text) that their enzyme is 'oxidized' or 'reduced'.

Line 157 retains some unfortunate word use, which another reviewer also flagged. the StnABC protomers cannot be called monomers. This is incorrect, since they are heterotrimers of the monomers StnA, StnB and StnC. Please consider use of the word 'protomer' to describe the functional StnABC unit (and define this terminology). Then the inconsistent sentences become: 'The observed complex consisted of four heterotrimeric protomers', wherein the tetrameric core was composed of the StnC monomers of each of the 4 protomers. Each of the StnC monomers was in turn flanked by a more dynamic.... This proposed notation is already in use in line 171.

Lines 183-187 provide important information on the working of StnABC and its relationship with HydABC, thank you for this addition. However for improved clarity I suggest:

'...also reported for the electron bifurcating hydrogenase HydABC, which displays increased bifurcating activity when exogenous FMN is provided, suggesting that activity displayed in the absence of exogenous FMN is attributable to a fraction of the population that retains FMN that co-purified with the protein, despite modest binding affinity.

The use of 'apo' is mis-leading in this paper. It indicates to readers that cofactors have been removed or are absent, when in fact there is likely a sub-stoichiometric population of FMN present, and full Fe

occupancy as well as full FAD occupancy. Thus 'apo' at the very least needs to clarify that it is only partially apo- and only refers to FMN. I recommend use of the 'as-isolated' nomenclature for the samples that are as-isolated (sub-stoichiometric FMN) to solve this issue. Samples augmented by FMN and NADPH can be so named also, using (for example) aug(FMN, NADPH) or a short version defined at time of first use such as Stn[^]FNH (^ means superscript).

Recall that FMN and FAD are cofactors, not substrates, and the nicotinamides are substrates/products of this enzyme. We actually do not know what oxidation state of NADP/H may remain bound.

Lines 188-190 are incomprehensible. Please rephrase. Perhaps two short sentences can be clearer than one long one? I cannot offer suggestions because I am not sure what message is intended.

Lines 237-238, and 394-395. The interesting phenomenon raised by the authors extends even to bifurcating ETFs. Please see for example Mohamed Raseek et al (2022) <https://doi.org/10.1016/j.jbc.2022.101733>

Lines 316-317 are internally inconsistent and redundant: 'stabilizing a flavo semiquinonic state of flavin instead of ASQ'. I suggest:

'stabilizing a hydroquinonic state of flavin instead of ASQ' if I have correctly understood the intent.

Discussion leads with considerable descriptive material on Stn that should have been much more valuable to readers if provided in the introduction, for example domain homologies. Moreover crucial foundational literature on this topic has not been cited. Please provide references to early determinations of strong homologies between Stn domains and domains of other redox enzymes. This information will allow the current new structural work to be correctly placed within a growing body of knowledge and the efforts of the giants upon whose shoulders we stand. Failure to provide this lineage may also suggest that the authors are not good scholars, which would be most unfortunate.

Meanwhile the discussion will be much more potent if it leads with the new conclusions presented, starting in line 374. This finding, which is genuinely structural (like the data) provides a nice segue to the mechanistic proposals.

Reviewer #2 (Remarks to the Author):

The authors have made substantial improvements to the manuscript and adequately responded to my criticisms/comments. I can now recommend publication of the current manuscript.

Reviewer #4 (Remarks to the Author):

The authors have not properly addressed the concerns we raised in their revised manuscript.

Major concerns:

1. We asked the clarification of the sample states (see previous major concerns #1). The authors responded by changing the names of the states. However, the naming of “Apo” and “Holo” still does not make sense and potentially more confusing. An apoenzyme has no redox cofactors (or prosthetic groups) by definition. That is not what the authors mean. Their “apo” form is the enzyme with all redox cofactors except FMN. They did not simply add FMN to reconstitute what should then be the “holoenzyme”. They added FMN (a redox cofactor) together with NADPH and NAD, which are substrates (Table 3). So, neither their definition of “apo” and “holo” is correct and will confuse the readers. They could simply name state 1 of the enzyme “as purified” with depleted FMN, and state 2 of the enzyme “as purified+FMN+NADPH+NAD” and not “apo” and “holo”.

2. We appreciate the somewhat improved modeling for NADP. However, the authors did not show the local density around all co-factors (See the previous major concerns #4). We understand that local density quality may not be good enough for atomic model building in certain partially flexible regions. However, the fitting between the model and the EM density around each cofactor/ligand must be shown in supplementary figures at both low and high display thresholds, so the readers can understand the quality of the models and know what’s experimentally determined and what’s merely suggestion/modeling.

3. The authors have not modified the misleading claim in Discussion (see the previous minor concerns #4). We agree that their work provide structural evidence on the “versatility of the BC module”, as they note in their reply. But their claim is wrong for two reasons: 1) The modules that couple to the electron bifurcation module are entirely different in [NiFe]-HydABC SL and [FeFe]-HydABC, and this already demonstrates structural diversity of the BC module; 2) Ref 20 has shown that a broad array of metabolic processes is coupled to the BC module through phylogenomic analysis. So, the functional versatility has been demonstrated.

Minor concerns:

1. We feel obligated to point out serious issues with the statement about electron bifurcation in the authors' reply to reviewers 1 and 2. The authors claim that the electron bifurcation in complex III is achieved by conformational change of the Rieske domain and explain the mechanism of HydABC-like enzymes using the Q-cycle as example. However, this is not what we understand about the electron bifurcation in the Q-cycle. The electron bifurcation there is achieved by quinol/quinone (see 2001 Trends Biochem Sci 26, 445-51; Yuly et al. 2019, Chem Commun 55, 11823-11832). The redox midpoint potential of Rieske FeS cluster can vary depending on the local environment, but not to an extent that would generate two low potential electrons (see Fig. 2c in the 2001 review). So, the theory of the electron bifurcation in the authors' previous paper is different from the well-established mechanism, and their proposed mechanism is highly speculative. It is wrong to claim that the mechanism in FeFe-HydABC resembles the Q-cycle.

2. The references authors mentioned are not the ones in the manuscript. In their reply to reviewer #2: "Feng et. al. (ref 18)". Ref18 in the manuscript is Furlan et al paper.

Reviewer #1 (Remarks to the Author):

The results are certainly noteworthy and will be of significance to the field. The work is original and compares favorably with existing literature. Arguments, experimental support and methodology are sound, but chemical units are needed in the table that reports catalytic activities.

Methods and description of samples has been significantly improved in this revision. Several ambiguities and mis-statements have been corrected with the result that clarity is significantly improved.

Thank you for the effort this entailed.

This is much improved, so that even aspects that were good before, are able to shine. I note for example the glorious figures and the numerous assays of catalytic activity, performed in triplicate, which are such excellent complements to structural findings.

We again express our gratitude and thank the referee for the positive assessment of our revised work. We have addressed all the minor points below.

HOWEVER: no chemical units were to be found for any of the assays, and the Table 1 reports wavelengths at which observations were made in lieu of units. The wavelengths describe how the assay was performed, but they do not allow any quantitative conclusions to be made with the numbers in the table. Only comparisons between protein variants. Please provide SOMEWHERE the conversion between 'Units' and $\mu\text{moles/L}$ ($=\mu\text{M}$) limiting substrate consumed per $\mu\text{mole/L}$ concentration of enzyme per unit of time (presumably at saturating concentrations of all other co-substrates). If more than one (co)substrate is limiting then there will be additional units, but they should all be units of concentration ($\mu\text{moles/mL}$). Ideally, the enzyme concentrations should be in some molar quantity not mg/mL , so they provide chemical insight. Sadly the biochemical community fails often in this regard and the consequence is that readers do not realize just how good our favorite catalysts really are!

Thank you for pointing it out, we have now added the units in the supplementary table 1.

Several abbreviations are still not defined anywhere: GlTD, NuoE..... at least provide provide a glossary in the SI. Only insiders and nerds realize that modules of additional NAD(P)H-utilizing enzymes are implied, so this is not merely a good habit, it is required for readers to appreciate the significance of the author's finding of homology with domains of StnC. Lines 203-204 did a very good job in this regard.

All the missing abbreviations have now been added to the manuscript.

Thank you for providing critical information on stoichiometry of bound cofactors, this greatly increases the value of the results. At the time of first reporting, please tell the reader the number of Fe expected (42) instead of simply saying 'as expected', which omits the information needed for readers to assess your results (line 124).

We have now revised the sentence.

Instead of the term 'oxidized', please call the untreated enzyme 'as-isolated'. No experimental determination has been made of oxidation state, so the vocabulary should avoid making unsubstantiated claims. Moreover this will remove an apparent claim to have oxidized the samples and maintained them as such during irradiation with a stream of electrons (!!). Since the samples were never actively oxidized (not even exposed to air), and there is not yet an agreed-upon way to keep these samples oxidized during EM, even if they started that way (which I very much doubt), the authors should not claim to have 'oxidized' material, anywhere.

Similarly, since there is no determination of the oxidation/reduction state, replace the terminology

'reduced' with '+NADPH' This is a simple accurate statement that will be clear and honest. It completely obviates any responsibility to determine oxidation states and it allows the reader to know what is present in the samples.

Thank you for the suggestion. In order to better clarify the states of the two structures, we have omitted the word apo/olo or oxidized/reduced from the manuscript. Instead, we describe the two states as; StnABC_{S1} and StnABC_{S2}.

StnABC_{S1} is the structure of the isolated or purified state of the enzyme in the absence of any added substrates (NADPH, NAD⁺, and Fd) or cofactors (FMN).

StnABC_{S2} is the structure of the purified enzyme with the addition of exogenous cofactor (FMN) and substrates (NADPH, NAD⁺, and Fd).

Of note, the cofactor FAD was added to both states during purification (see methods).

The authors have convinced me that spectroscopy to make determinations of oxidation states is too much to ask, I agree, ON THE CONDITION that the authors cease making claims (throughout the text) that their enzyme is 'oxidized' or 'reduced'.

The states of the enzymes have now been reassigned, and therefore we have omitted the usage of the words 'oxidized' or 'reduced' to describe the states.

Line 157 retains some unfortunate word use, which another reviewer also flagged. the StnABC protomers cannot be called monomers. This is incorrect, since they are heterotrimers of the monomers StnA, StnB and StnC. Please consider use of the word 'protomer' to describe the functional StnABC unit (and define this terminology). Then the inconsistent sentences become: 'The observed complex consisted of four heterotrimeric protomers', wherein the tetrameric core was composed of the StnC monomers of each of the 4 protomers. Each of the StnC monomers was in turn flanked by a more dynamic.... This proposed notation is already in use in line 171.

Thanks for pointing this out. We have now revised the sentence.

Lines 183-187 provide important information on the working of StnABC and its relationship with HydABC, thank you for this addition. However for improved clarity I suggest: '...also reported for the electron bifurcating hydrogenase HydABC, which displays increased bifurcating activity when exogenous FMN is provided, suggesting that activity displayed in the absence of exogenous FMN is attributable to a fraction of the population that retains FMN that co-purified with the protein, despite modest binding affinity.

We thank the referee for the suggestion. We have now included the suggested part in the manuscript.

The use of 'apo' is mis-leading in this paper. It indicates to readers that cofactors have been removed or are absent, when in fact there is likely a sub-stoichiometric population of FMN present, and full Fe occupancy as well as full FAD occupancy. Thus 'apo' at the very least needs to clarify that it is only partially apo- and only refers to FMN. I recommend use of the 'as-isolated' nomenclature for the samples that are as-isolated (sub-stoichiometric FMN) to solve this issue. Samples augmented by FMN and NADPH can be so named also, using (for example) aug(FMN, NADPH) or a short version defined at time of first use such as Stn[^]FNH (^ means superscript).

Recall that FMN and FAD are cofactors, not substrates, and the nicotinamides are substrates/products of this enzyme. We actually do not know what oxidation state of NADP/H may remain bound.

As mentioned in the above point, we have removed the word apo or holo from the manuscript and instead used StnABC_{S2} and StnABC_{S1} to describe the individual states obtained with and without added cofactors and substrates, respectively.

Lines 188-190 are incomprehensible. Please rephrase. Perhaps two short sentences can be clearer than one long one? I cannot offer suggestions because I am not sure what message is intended.

We have rephrased the sentence for better clarity.

Lines 237-238, and 394-395. The interesting phenomenon raised by the authors extends even to bifurcating ETFs. Please see for example Mohamed Raseek et al (2022) <https://doi.org/10.1016/j.jbc.2022.101733>

Thank you, we have now included this reference at the requested place in the manuscript (reference 23 in revised manuscript).

Lines 316-317 are internally inconsistent and redundant: 'stabilizing a flavo semiquinonic state of flavin instead of ASQ'. I suggest: 'stabilizing a hydroquinonic state of flavin instead of ASQ' if I have correctly understood the intent.

Thanks for pointing this out, we have now rephrased the sentence.

Discussion leads with considerable descriptive material on Stn that should have been much more valuable to readers if provided in the introduction, for example domain homologies. Moreover crucial foundational literature on this topic has not been cited. Please provide references to early determinations of strong homologies between Stn domains and domains of other redox enzymes. This information will allow the current new structural work to be correctly placed within a growing body of knowledge and the efforts of the giants upon whose shoulders we stand. Failure to provide this lineage may also suggest that the authors are not good scholars, which would be most unfortunate.

We would like to point out that all the important literature describing the homology of Stn subunits with other enzymes has been cited. Moreover, the first Stn paper describing its discovery and the modular nature of its subunits is cited (see references 12, 17, and 22 in the main text).

Meanwhile the discussion will be much more potent if it leads with the new conclusions presented, starting in line 374. This finding, which is genuinely structural (like the data) provides a nice segue to the mechanistic proposals.

Thank you for the suggestion, we have now moved the requested paragraph at the end. Moreover, we have also toned down our claim that the StnABC complex is the first structure to demonstrate the modular nature of the HydBC or StnAB subcomplex.

Reviewer #2 (Remarks to the Author):

The authors have made substantial improvements to the manuscript and adequately responded to my criticisms/comments. I can now recommend publication of the current manuscript.

We thank the referee for praising our revised work and recommending it for publication in Nature Communications.

Reviewer #4 (Remarks to the Author):

The authors have not properly addressed the concerns we raised in their revised manuscript.

We thank the referee for providing constructive comments and suggestions. We have addressed all the points thoroughly.

Major concerns:

1. We asked the clarification of the sample states (see previous major concerns #1). The authors responded by changing the names of the states. However, the naming of "Apo" and "Holo" still does not make sense and potentially more confusing. An apoenzyme has no redox cofactors (or prosthetic groups) by definition. That is not what the authors mean. Their "apo" form is the enzyme with all redox cofactors except FMN. They did not simply add FMN to reconstitute what should then be the "holoenzyme". They added FMN (a redox cofactor) together with NADPH and NAD, which are substrates (Table 3). So, neither their definition of "apo" and "holo" is correct and will confuse the readers. They could simply name state 1 of the enzyme "as purified" with depleted FMN, and state 2 of the enzyme "as purified+FMN+NADPH+NAD" and not "apo" and "holo".

We thank the referee for pointing this out. Taking into consideration the comments from referee 1 as well, we have now decided to omit the word apo and holo from the manuscript. Also, agreeing with referee's suggestion, we have now described the two states as; StnABC_{S1} and StnABC_{S2}.

StnABC_{S1} is the structure of the isolated or purified state of the enzyme in the absence of any added substrates (NADPH, NAD⁺, and Fd) or cofactor (FMN).

StnABC_{S2} is the structure of a purified enzyme with added exogenous cofactor (FMN) and substrates (NADPH, NAD⁺, and Fd).

Of note, the cofactor FAD was added to both states during purification (see methods).

2. We appreciate the somewhat improved modeling for NADP. However, the authors did not show the local density around all co-factors (See the previous major concerns #4). We understand that local density quality may not be good enough for atomic model building in certain partially flexible regions. However, the fitting between the model and the EM density around each cofactor/ligand must be shown in supplementary figures at both low and high display thresholds, so the readers can understand the quality of the models and know what's experimentally determined and what's merely suggestion/modeling.

We have now revised the Supplementary Fig. 4. The cryo-EM density for all the cofactors and clusters is now shown. For NADPH and the FeS clusters in N- and C-ter of the StnB subunit are shown at low and high thresholds. We have also mentioned in the caption that NADHP modelling is now performed based on the crystal structure of NfnAB bound to NADPH.

3. The authors have not modified the misleading claim in Discussion (see the previous minor concerns #4). We agree that their work provide structural evidence on the "versatility of the BC module", as they note in their reply. But their claim is wrong for two reasons: 1) The modules that couple to the electron bifurcation module are entirely different in [NiFe]-HydABC_{SL} and [FeFe]-HydABC, and this already demonstrates structural diversity of the BC module; 2) Ref 20 has shown that a broad array of metabolic processes is coupled to the BC module through phylogenomic analysis. So, the functional versatility has been demonstrated.

We apologise to the referee for this oversight. We never wanted to claim that StnABC is the first structure to demonstrate the modular nature of the HydBC/StnAB subcomplex. We have now toned down this claim and have made the necessary corrections in the discussion.

Minor concerns:

1. We feel obligated to point out serious issues with the statement about electron bifurcation in the authors' reply to reviewers 1 and 2. The authors claim that the electron bifurcation in complex III is achieved by conformational change of the Rieske domain and explain the mechanism of HydABC-like enzymes using the Q-cycle as example. However, this is not what we understand about the electron bifurcation in the Q-cycle. The electron bifurcation there is achieved by quinol/quinone (see 2001 Trends Biochem Sci 26, 445-51; Yuly et al. 2019, Chem Commun 55, 11823-11832). The redox midpoint potential of Rieske FeS cluster can vary depending on the local environment, but not to an extent that would generate two low potential electrons (see Fig. 2c in the 2001 review). So, the theory of the electron bifurcation in the authors' previous paper is

different from the well-established mechanism, and their proposed mechanism is highly speculative. It is wrong to claim that the mechanism in FeFe-HydABC resembles the Q-cycle.

We would like to clarify that we do not claim the mechanism of HydABC to resemble the Q-cycle. However, the concept of redox-driven conformational changes as an alternative to kinetically gate electron transfer, as used by the HydC (or StnA) subunit, is similar to the one employed by the Rieske domain. At the same time, we agree that our mechanism is speculative, however, we have clearly shown that the well-established definition of flavin-based electron bifurcation cannot be used for describing bifurcation in HydABC or StnABC or Ni-Fe HydABC_{SL}.

2. The references authors mentioned are not the ones in the manuscript. In their reply to reviewer #2: "Feng et. al. (ref 18)". Ref18 in the manuscript is Furlan et al paper.

This reference has now been corrected in the revised manuscript.